# ThoughtFold: Folding Reasoning Chains via Introspective Preference Learning

Ziyan Liu [* 1 2]  Xueda Shen [* 1]  Yuzhe Gu [* 1]  Songyang Gao [1]  Kuikun Liu [1]  Guangran Cheng [1]  Chengqi Lyu [1]
Dahua Lin [1]  Wenwei Zhang [1 †]  Kai Chen [1 †]

## Abstract

Large Reasoning Models (LRMs) have achieved remarkable progress thanks to Reinforcement Learning with Verifiable Rewards (RLVR) on Chain-of-Thoughts (CoTs). However, since long CoTs naturally contain trial and errors and mainstream RLVR approaches choose outcome-correct CoT trajectories for memorization, the *redundant explorations* in long CoTs are inevitably reinforced, which results in the over-thinking issues of LRMs. Previous attempts to resolve this issue mainly give more advantage to shorter trajectories, yet their learning signals are still outcome-based and cannot reduce the memorization of redundant explorations in long CoTs. Therefore, we propose **ThoughtFold**, a framework that leverages fine-grained preference learning to mitigate redundant explorations for efficient reasoning. ThoughtFold employs an introspective strategy to identify redundancy within each correct trajectory, which yields a spectrum of candidate sub-trajectories. Leveraging this spectrum, we introduce a masked preference optimization objective that explicitly penalizes redundant explorations and encourages the model to directly bridge essential reasoning segments, effectively folding its reasoning chains into a more concise path. Extensive experiments show that ThoughtFold significantly enhances efficiency. It reduces the token usage of DeepSeek-R1-Distill-Qwen-7B by approximately 56% while maintaining state-of-the-art accuracy.

## 1. Introduction

Solving complex problems with reasoning capability forms one of the cornerstones of human cognition, which is also an important component in building Artificial General Intelligence (AGI) (Weston et al., 2015; Yang et al., 2018; Hendrycks et al., 2021a). Recently, Large Reasoning Models (LRMs) have made significant progress in mimicking human thinking by using Reinforcement Learning with Verifiable Rewards (RLVR) on Chain-of-Thoughts (CoTs), which treats the correctness of CoTs as the optimization target (Wei et al., 2023). However, despite the success, LRMs suffer from "overthinking", *i.e.* these models tend to generate lengthy CoTs that interweave necessary logical deductions with redundant exploration such as self-repetition or off-target attempts (Chen et al., 2025; Sui et al., 2025).

The overthinking phenomenon is an algorithmic artifact of RLVR. Because RLVR supervises models based solely on the final correctness of the trajectory, which naturally involves hit-or-miss exploration, it indiscriminately reinforces all steps within a correct trajectory. Consequently, the model memorizes both the necessary deductions (signal) and the redundant explorations (noise), learning to overthink simultaneously when developing reasoning abilities (Zelikman et al., 2022; Uesato et al., 2022; Chen et al., 2025). Although previous attempts have tried to mitigate this by introducing a reward factor to penalize the trajectory length (Arora & Zanette, 2025a; Li et al., 2025; Yi & Wang, 2025), such trajectory-level penalties cannot achieve step-level credit assignment, limiting their supervision and mitigation of redundant exploration in inference trajectories.

To address this limitation, we propose ThoughtFold, which integrates outcome-based RLVR with fine-grained preference learning for efficient reasoning. As shown in Figure 1, unlike vanilla RLVR strategies that uniformly reinforce all steps in a correct trajectory, our method performs fine-grained preference learning by identifying and explicitly fold redundant thoughts. Specifically, ThoughtFold employs an int rospective strategy for redundancy identification. Starting with an outcome-correct trajectory, we iteratively remove specific reasoning segments to verify if the model can still derive the correct answer. This process yields a spectrum of trajectories, distinguishing between concise successes (where redundancy is successfully removed) and over-simplified failures (where essential logic is broken). Based on this spectrum, ThoughtFold applies a mask-based fine-grained preference optimization to ex-

---

[*]Equal contribution  [1]Shanghai Artificial Intelligence Laboratory [2]University of Science and Technology of China. Correspondence to: Wenwei Zhang <zhangwenwei@pjlab.org.cn>, Kai Chen <chenkai@pjlab.org.cn>.

*Proceedings of the $43^{rd}$ International Conference on Machine Learning*, Seoul, South Korea. PMLR 306, 2026. Copyright 2026 by the author(s).

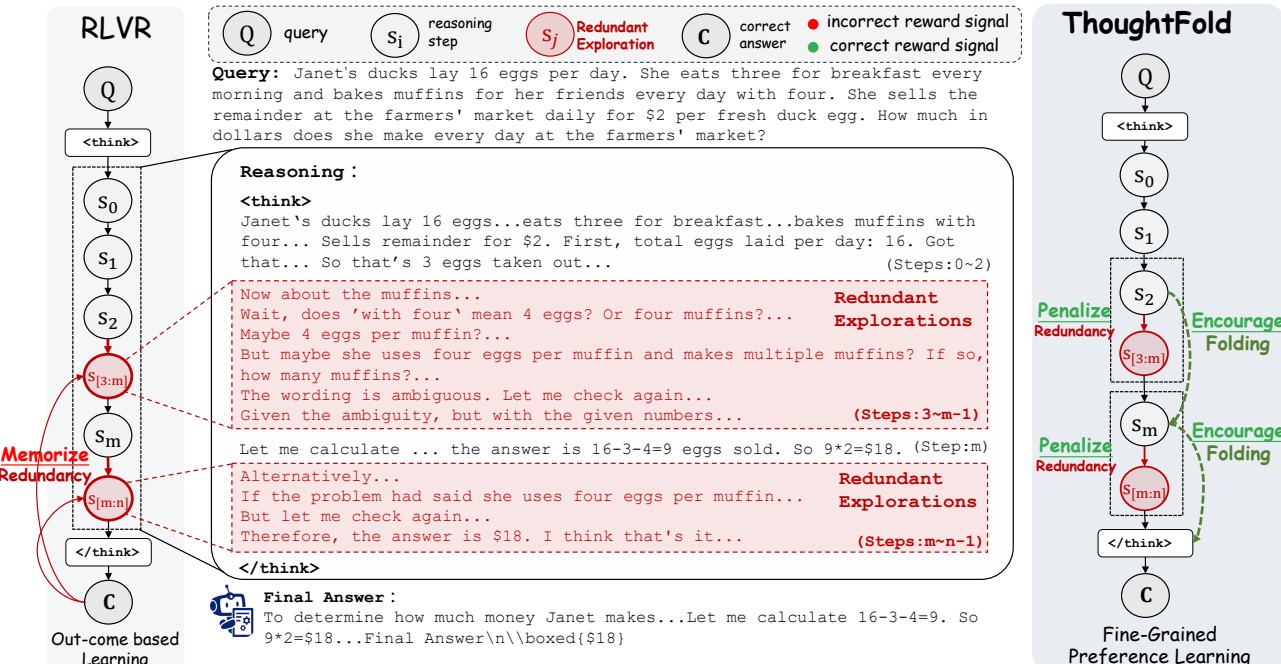

*Figure 1.* **Comparison between RLVR and ThoughtFold.** An outcome-correct CoT (middle) naturally contains redundant explorations. RLVR (left) memorizes these steps by uniformly reinforcing the entire CoT. In contrast, ThoughtFold (Right) identifies and penalizes redundant steps, folding the reasoning chain by encouraging direct bridging between essential reasoning segments.

plicitly penalize the identified redundant explorations and encourage the model to directly bridge the essential logical steps.

Through extensive experiments with Qwen3 and DeepSeek-series reasoning models on benchmarks ranging from GSM8K to AIME and GPQA, we demonstrate that Thought-Fold significantly enhances reasoning efficiency. Notably, it reduces the average token consumption of DeepSeek-R1-Distill-Qwen-7B by approximately 56% while maintaining state-of-the-art accuracy, surpassing recent efficient reasoning works. Further analysis confirms that this efficiency improvement stems from a change in reasoning behavior. ThoughtFold introduces a more concentrated reasoning mode that mitigates structural redundancy rather than simply reshaping the output length distribution.

## 2. Preliminaries

### 2.1. Formulation

Let $\pi_\theta$ denote a Large Reasoning Model (LRM) parameterized by $\theta$. Given an input query $x$, the model generates a response trajectory $\tau$, which is composed of an intermediate chain-of-thought (CoT) reasoning $z$ (typically enclosed within <think> tags) and a final answer $y$, denoted as $\tau = (z, y)$. We decompose the reasoning trajectory $z$ as a sequence of discrete reasoning steps $z = \{s_1, s_2, \ldots, s_N\}$ using predefined rules (*e.g.*,"\n\n") (Zhang et al., 2025), where each $s_i$ represents a distinct logical unit within the

reasoning path. Such decomposition transforms the continuous reasoning stream into discrete units, enhancing the management and cognition of the reasoning process.

### 2.2. Reinforcement Learning for LLMs

**RLVR.** Current Large Reasoning Models (LRMs) predominantly adopt the Reinforcement Learning with Verifiable Rewards (RLVR) paradigm to scale reasoning capabilities (DeepSeek-AI et al., 2025; Yu et al., 2025). Formally, RLVR aims to maximize the expected reward of generated trajectories:

$$\max_\theta \mathcal{J}(\theta) = \mathbb{E}_{x \sim \mathcal{D}, \tau \sim \pi_\theta(\cdot|x)}[r(y)], \quad (1)$$

where $r(y)$ is a binary outcome-based reward. To optimize this, policy gradient methods estimate the gradient using an advantage term $A_t$:

$$\nabla_\theta \mathcal{J}(\theta) = \mathbb{E}_{x \sim P(X), \tau \sim \pi_\theta} \left[ \sum_{t=1}^{|\tau|} A_t \nabla_\theta \log \pi_\theta(\tau_t | x, \tau_{<t}) \right] \quad (2)$$

In representative algorithms like GRPO (DeepSeek-AI et al., 2025), this advantage is estimated by normalizing rewards within a group of $G$ sampled outputs $\{y^1, \ldots, y^G\}$ for the same prompt:

$$A_t^{(i)} = \frac{r(y^i) - \text{mean}(\{r(y^i)\}_{i=1}^G)}{\text{std}(\{r(y^i)\}_{i=1}^G) + \epsilon}. \quad (3)$$

Crucially, for a given trajectory $i$, the advantage $A_t^{(i)}$ is assigned uniformly to every token in the chain-of-thought $z$. Since the calculation relies solely on the final answer correctness, this coarse-grained signal inadvertently reinforces redundant steps $s_t$ contained within a correct trajectory.

**Direct Preference Optimization.** Direct Preference Optimization (DPO) (Rafailov et al., 2024) aligns the policy $\pi_\theta$ using preference data $\mathcal{D} = \{(x, y_w, y_l)\}$, where the winner $y_w$ denotes the preferred response with higher quality over loser $y_l$ ($y_w \succ y_l$). By defining the implicit reward $r_\theta(x, y) = \beta \log \frac{\pi_\theta(y|x)}{\pi_{\text{ref}}(y|x)}$, DPO objective is formulated as:

$$\mathcal{L}_{\text{DPO}}(\theta) = -\mathbb{E}_{(x,y_w,y_l)\sim\mathcal{D}}\left[\log\sigma\left(r_\theta(x,y_w) - r_\theta(x,y_l)\right)\right] \quad (4)$$

where $\sigma$ is the sigmoid function. Mask-DPO (Gu et al., 2025) extends this by isolating fine-grained supervision signals via sequence decomposition. Given step-level annotations $a = \{a_i\}$ where $a_i = 0$ (where 0 and 1 denote correctness and error, respectively), Mask-DPO replaces the global reward $r_\theta$ with masked divergences $\mathcal{M}$. These terms specifically reinforce correct steps in $y_w$ and penalize hallucinations in $y_l$:

$$\mathcal{M}_w(x, y_w, a_w) = \sum_i \mathbb{I}(a_i^w = 0) \log \frac{\pi_\theta(s_i^w|x, s_{<i}^w)}{\pi_{\text{ref}}(s_i^w|x, s_{<i}^w)},$$
$$\mathcal{M}_l(x, y_l, a_l) = \sum_j \mathbb{I}(a_j^l = 1) \log \frac{\pi_\theta(s_j^l|x, s_{<j}^l)}{\pi_{\text{ref}}(s_j^l|x, s_{<j}^l)}, \quad (5)$$

The optimization objective is then reformulated to maximize the margin between these masked terms:

$$\mathcal{L}_{\text{MDPO}}(\theta) = -\mathbb{E}_\mathcal{D}\big[\log\sigma\big(\beta\mathcal{M}_w(x, y_w, a_w) - \beta\mathcal{M}_l(x, y_l, a_l)\big)\big]. \quad (6)$$

This formulation forces the model to specifically learn the fine-grained correct signal, preventing the reinforcement of errors embedded in preferred responses or the suppression of correct segments in rejected ones.

## 3. Method

We propose ThoughtFold, a reinforcement learning framework for efficient reasoning, as shown in Figure 2, which integrates (a) fine-grained preference learning with (b) outcome-based RLVR. For fine-grained preference learning, ThoughtFold adopts an *introspective strategy* to identify redundant explorations and construct preference pairs (Sec. 3.1). Then it employs a carefully designed *dynamic mask strategy* for fine-grained reasoning policy optimization (Sec. 3.2), which provides step-level signals to explicitly mitigate redundant exploration. We present the joint optimization target of ThoughtFold in Sec. 3.3.

### 3.1. Introspective Redundancy Identification

We use an introspective strategy to detect redundancy within correct trajectories. Given an input query $x$ and a verified trajectory $\tau_{ref} = (z_{ref}, y^*)$ generated by LRM $\pi_\theta$, where the reasoning chain $z_{ref} = \{s_1, \ldots, s_n\}$ leads to the correct answer $y^*$, we adopt a prune-and-verify strategy to iteratively identify redundancy within $\tau_{ref}$. In each iteration $j$, we prune a subset of reasoning steps in $z_{ref}$ to construct a shortened candidate reasoning chain $z_{cand,j}$. Then we force the model $\pi_\theta$ to immediately generate an answer $\hat{y}_{cand,j} \sim \pi_\theta(\cdot|x, z_{cand,j})$ based on the compressed CoT $z_{cand,j}$ by appending an end-of-thought token (e.g., `</think>`). We adopt an iterative search to adaptively adjust the pruning ratio of reasoning steps based on the correctness of $\hat{y}_{cand,j}$. The search process is structured into two phases to filter out *self-repetition* and *off-target attempts*:

**Phase 1: Tail Truncation.** Models often continue generating tokens even after the logical conclusion has been reached (self-repetition). Therefore, we perform a binary search on the prefix length $m_j$. In each iteration, we construct the candidate reasoning trajectory $z_{cand,j}$ by truncating the refinal sequence at the step level: $z_{cand,j} = z_0[: m_j]$.

**Phase 2: Internal Folding.** Given the shortest valid reasoning chain $z_{trunc}$ from (1), we proceed to eliminate internal redundant steps (off-target attempts). We assess the importance of each step $s_t$ based on the model's intrinsic attention distribution. The step-level importance score $I(s_t)$ is calculated by averaging the attention weights from generated answer tokens $y_{trunc}$ to the tokens within step $s_t$:

$$I(s_t) = \frac{1}{|s_t|}\sum_{u\in s_t}\left(\frac{1}{|y_{trunc}|}\sum_{v\in y_{trunc}}\mathcal{A}_{v\to u}\right), \quad (7)$$

where $\mathcal{A}_{v\to u}$ denotes the attention weight from answer token $v \in y_{trunc}$ to reasoning token $u \in z_{trunc}$. Based on these scores, we perform a binary search on the retention ratio $k_j$. We construct $z_{cand,j}$ by folding away the low-utility steps and bridging the remaining top-$k_j\%$ steps that possess the highest $I(s_t)$. The importance score computation is required once for each correct sample. We provide justification of attention-based importance score in Appendix E.

We illustrate this two-stage binary search process in Figure 2. The detailed implementation is outlined in Appendix C.

**Preference Pair Construction.** This introspective process yields a spectrum of candidate sub-trajectories derived from $z_{ref}$. Leveraging this spectrum, we dynamically construct a preference dataset $\mathcal{D}$. We maintain a $z_{best}$ (initialized as $z_{ref}$) to represent the shortest valid trajectory discovered so far. For each candidate $z_{cand,j}$ sampled during searching, we update $\mathcal{D}$ based on the outcome correctness:

**(1) Concise Success** ($z_{cand,j} \succ z_{best}$)**.** If $z_{cand,j}$ derives a

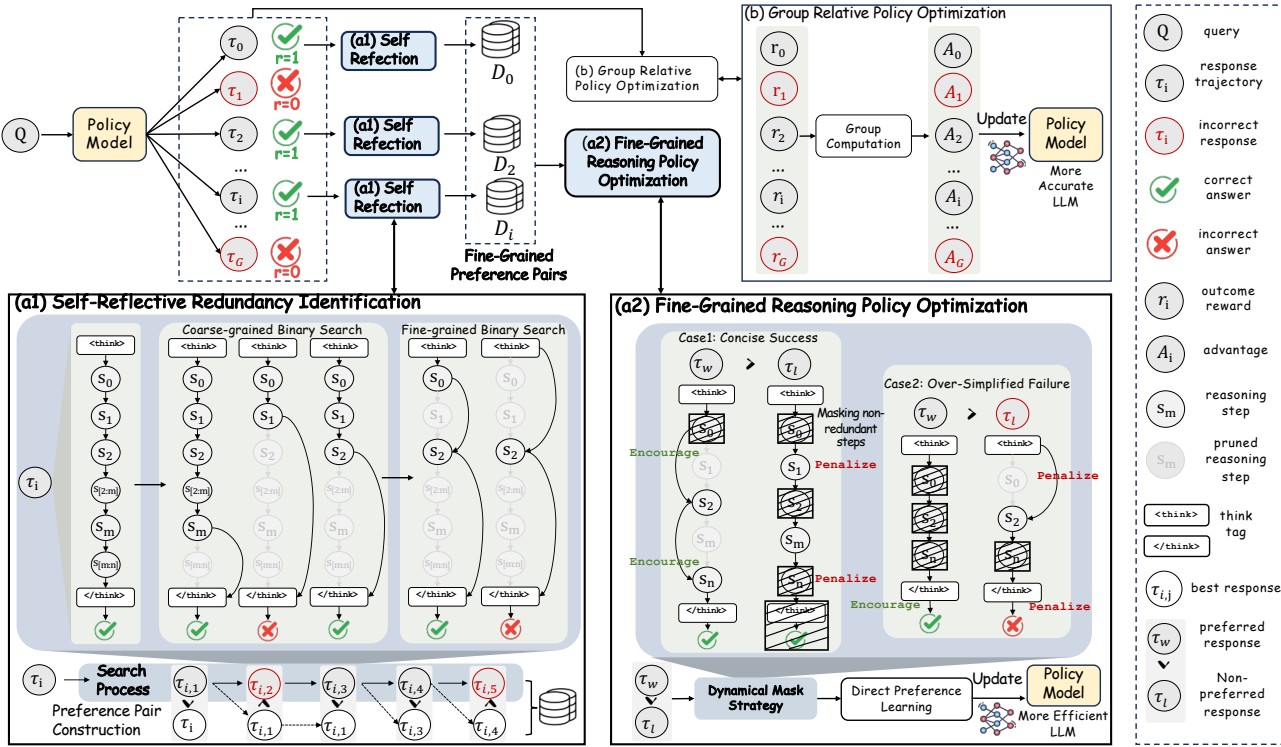

*Figure 2.* **Overview of ThoughtFold.** The framework integrates outcome-based reinforcement (e.g., GRPO) with fine-grained preference learning. We employ an introspective strategy to iteratively identify reasoning redundancies. These refined trajectories form preference pairs, explicitly aligning the model toward correct and concise reasoning paths.

correct answer, it represents a more concise reasoning path. We append this positive instance to our dataset: $\mathcal{D} \leftarrow \mathcal{D} \cup \{(z_{cand,j}, z_{best})\}$, and update the anchor $z_{best} \leftarrow z_{cand,j}$.

**(2) Over-Simplified Failure** ($z_{best} \succ z_{cand,j}$). If $z_{cand,j}$ fails to produce the correct answer, it indicates that the pruning was too aggressive and severed essential logic. We record as a negative constraint: $\mathcal{D} \leftarrow \mathcal{D} \cup \{(z_{best}, z_{cand,j})\}$.

### 3.2. Fine-Grained Reasoning Policy Optimization

Based on the constructed dataset $\mathcal{D}$, we perform fine-grained preference learning for concise reasoning. While standard DPO effectively aligns model outputs, it applies a global penalty to the entire rejected trajectory. This indiscriminately punishes essential reasoning steps within rejected trajectories. To achieve fine-grained reasoning policy optimization, we adopt a *Dynamic Mask Strategy* to apply precise, step-level signals on each trajectory. This strategy explicitly penalize redundant explorations and encourages the model to directly bridge the essential logical steps, folding its reasoning chain into an efficient path.

**Dynamic Mask Strategy.** For clarity, we introduce the concept of the **Fold Anchor**. We define the Fold Anchor as the specific reasoning step in the shortened trajectory $z_{cand,j}$ that immediately follows the segment pruned from $z_{best}$.

Conceptually, this step serves as the bridge that reconnects the logical flow after removing redundant explorations. For example, in Figure 2, consider a sequence $\{s_0, s_1, s_2\}$ in $z_{best}$. If the redundant step $s_1$ is folded away, the subsequent step $s_2$ preserved in $z_{cand,j}$ becomes the **Fold Anchor**.

For each preference pair in $\mathcal{D}$, we dynamically construct the step-level binary masks $(M_w, M_l)$:

**(1) Concise Success** ($z_{cand,j} \succ z_{best}$). In this scenario, the folded trajectory yields the correct answer with fewer steps, representing a successful logical shortcut. For the winner mask $M_w$, we activate the loss ($M_{w,t} = 1$) for Fold Anchors to encourage efficient connections between useful steps. For the loser mask $M_l$, we mask out ($M_{l,t} = 0$) the shared reasoning steps and the final correct answer, precisely penalizing redundant reasoning steps.

**(2) Over-Simplified Failure** ($z_{best} \succ z_{cand,j}$). Here, the shorter candidate fails to produce the correct answer, indicating that the folding is too aggressive and has severed essential logic. For $M_w$, we only encourage the correct answer ($M_{w,t} = 1$). For $M_l$, We penalize tokens within the Fold Anchor and the incorrect answer ($M_{l,t} = 1$). This discourages logical leaps that lack sufficient grounding.

**Fine-Grained Optimization Objective.** We incorporate these dynamic masks into the DPO reward to formulate the

masked implicit reward $\mathcal{M}(z, M)$ (see Eq. 8). By applying step-level masking, we precisely penalize redundant exploration while encouraging the efficient connections between useful reasoning steps (i.e., the Fold Anchors).

$$\mathcal{M}(z, M, \theta) = \beta \sum_{t=1}^{|z|} M_t \log \frac{\pi_\theta(z_t|x, z_{<t})}{\pi_{\text{ref}}(z_t|x, z_{<t})}. \quad (8)$$

Substituting this term into the DPO objective, we formulate the fine-grained reasoning policy optimization objective over $\mathcal{D}$ as follows:

$$\mathcal{J}_{\text{MDPO}}(\theta) = \mathbb{E}_{(x,z_w,M_w,z_l,M_l)\sim\mathcal{D}} \Big[ \log \sigma \big( \mathcal{M}(z_w, M_w, \theta) \\ - \mathcal{M}(z_l, M_l, \theta) \big) \Big]. \quad (9)$$

This objective guides the model to bypass redundant loops and directly bridge the logical gaps, resulting in a more compact and efficient reasoning policy.

### 3.3. Joint Optimization Target of ThoughtFold

The fine-grained optimization applies precise, step-level signals to enforce reasoning efficiency by identifying and penalizing redundancy. Since this optimization targets redundancy removal, many essential reasoning steps receive no direct supervision. We therefore incorporate the standard GRPO objective $\mathcal{J}_{\text{GRPO}}$ (as detailed in Section 2.2) to introduce trajectory-level supervision, ensuring that the model maintains high reasoning accuracy.

As shown in Figure 2, given a query $x$, we generate a group of $G$ trajectories. For each correct trajectory $\tau_i$ (the answer $y_i$ matches the ground truth), we construct a preference dataset $D_i$ along with corresponding masks for fine-grained policy optimization. These datasets are then concatenated to form an aggregated dataset $\mathcal{D}$. Then, the total optimization objective $\mathcal{J}_{\text{total}}$ is formulated as:

$$\max_\theta \mathcal{J}_{\text{total}}(\theta) = \mathcal{J}_{\text{GRPO}}(\theta) + \lambda \mathcal{J}_{\text{MDPO}}(\theta), \quad (10)$$

where $\lambda$ acts as a coefficient to balance the trade-off between reasoning accuracy and conciseness. The detailed iterative training process is presented in Algorithm 2.

## 4. Experiments

### 4.1. Experiment setup

**Models.** We evaluate ThoughtFold on four large reasoning models, including DeepSeek-R1-Distill-Qwen-7B, DeepSeek-R1-Distill-Qwen-14B (DeepSeek-AI et al., 2025), Qwen3-8B, and Qwen3-14B (Team, 2025). Despite achieving state-of-the-art performance on reasoning tasks, these models tend to generate overly verbose reasoning processes that contain excessive redundant information.

**Benchmarks.** To comprehensively assess the models' reasoning capabilities across diverse difficulty levels and domains, we select five popular benchmarks that reflect diverse levels of difficulty: **GSM8K** (Cobbe et al., 2021) serves as a foundational benchmark, comprising 1,319 high-quality test problems that demand rigorous multi-step reasoning. These problems are formulated as linguistically rich narratives rather than simple equations, requiring the model to decompose the query into several sequential arithmetic steps. To evaluate performance on more complex competition-level tasks, we utilize **MATH-500** (Hendrycks et al., 2021b), a challenging subset of 500 problems from the MATH dataset spanning algebra, geometry, and number theory. **AIME 2024** and **AIME 2025** (MAA Committees) comprise the latest problems from the American Invitational Mathematics Examination, serving as the standard benchmark for evaluating advanced reasoning and long-horizon planning capabilities. Moreover, to examine generalization beyond mathematics problems, we adopt **GPQA Diamond** (Rein et al., 2023) for evaluation. This dataset contains graduate-level questions in physics, chemistry, and biology, serving as a robust proxy for scientific reasoning capabilities where domain experts typically struggle.

**Metrics.** ThoughtFold is designed to improve correctness while minimizing inference length, thereby enabling more concise reasoning. We adopted two key metrics: Accuracy (i.e., pass@1) and Token Count (Tokens). Considering the inherent instability of generating long sequences, we performed 16 trials on AIME 2024 and AIME 2025, 8 trials on MATH-500 and GPQA Diamond, and 4 trials on GSM8K.

**Models.** We evaluate ThoughtFold on four large reasoning models, including DeepSeek-R1-Distill-Qwen-7B, DeepSeek-R1-Distill-Qwen-14B (DeepSeek-AI et al., 2025), Qwen3-8B, and Qwen3-14B (Qwen et al., 2025). Despite achieving state-of-the-art performance on reasoning tasks, these models tend to generate overly verbose reasoning processes that contain excessive redundant information.

**Baselines.** We benchmark ThoughtFold against representative RLVR frameworks to comprehensively evaluate both reasoning accuracy and efficiency. First, we establish the performance foundation using the **Vanilla** reasoning model and the standard **GRPO** (DeepSeek-AI et al., 2025). Second, we compare against **Length-Reward RL** methods, which incentivize conciseness via penalty terms. This category includes Short-RL (Yuan et al.) (building on Kimi k1.5 techniques (Team et al., 2025a)) and the recent RL + Length Penalty approach (Arora & Zanette, 2025b). Finally, we include **S-GRPO** (Dai et al., 2025), a state-of-the-art efficient reasoning method that utilizes serial grouping and decaying rewards to encourage early exits. We select S-GRPO as our primary advanced baseline as it demonstrates superior performance over inference-time acceleration strategies like

*Table 1.* Performance (Accuracy) and reasoning efficiency (Tokens) comparison across four models and five benchmarks.

| Method | GSM8K Acc | GSM8K Tokens | AIME 2024 Acc | AIME 2024 Tokens | AIME 2025 Acc | AIME 2025 Tokens | MATH-500 Acc | MATH-500 Tokens | GPQA Acc | GPQA Tokens | Overall Acc | Overall Tokens |
|---|---|---|---|---|---|---|---|---|---|---|---|---|
| *DeepSeek-R1-Distill-Qwen-7B* | | | | | | | | | | | | |
| Vanilla | 92.4 | 1,833 | 55.4 | 13,232 | 39.1 | 15,131 | 85.8 | 5,590 | 50.1 | 15,385 | 64.56 | 10,234 |
| GRPO | 93.2 | 1,767 | 55.0 | 13,451 | 39.0 | 14,926 | 93.6 | 5,317 | 50.7 | 15,817 | 66.30$_{+1.74}$ | 10,256$_{+0.2\%}$ |
| RL + Length Penalty | 92.4 | 1,062 | 51.9 | 7,464 | 35.5 | 9,976 | 92.2 | 2,451 | 49.1 | 3,984 | 64.22$_{-0.34}$ | 4,987$_{-51.3\%}$ |
| Short-RL | 93.1 | 1,102 | 53.7 | 7,239 | 35.2 | 9,779 | 91.7 | 2,234 | 49.3 | 3,897 | 64.60$_{+0.04}$ | 4,850$_{-52.6\%}$ |
| S-GRPO | 93.8 | 906 | 56.0 | 7,377 | – | – | 92.4 | 2,252 | 50.8 | 3,751 | – | – |
| ThoughtFold | 94.3 | 842 | 57.2 | 7,013 | 39.1 | 9,102 | 94.4 | 2,089 | 51.9 | 3,433 | **67.38**$_{+2.82}$ | **4,496**$_{-56.1\%}$ |
| *DeepSeek-R1-Distill-Qwen-14B* | | | | | | | | | | | | |
| Vanilla | 94.2 | 2,129 | 64.4 | 11,099 | 46.3 | 13,421 | 93.5 | 3,844 | 59.2 | 6,034 | 71.52 | 7,305 |
| GRPO | 95.3 | 2,120 | 65.8 | 13,504 | 47.3 | 13,519 | 84.0 | 4,471 | 58.9 | 7,354 | 70.26$_{-1.26}$ | 8,194$_{+12.2\%}$ |
| RL + Length Penalty | 94.7 | 775 | 55.0 | 7,950 | 43.2 | 9,289 | 92.4 | 1,993 | 56.0 | 4,380 | 68.26$_{-3.26}$ | 4,877$_{-33.2\%}$ |
| Short-RL | 95.1 | 781 | 61.7 | 7,561 | 42.0 | 9,279 | 93.1 | 2,021 | 57.3 | 4,241 | 69.84$_{-1.68}$ | 4,777$_{-34.6\%}$ |
| S-GRPO | 96.2 | 724 | 64.4 | 6,712 | – | – | 93.6 | 2,146 | 59.3 | 3,334 | – | – |
| ThoughtFold | 96.7 | 701 | 65.6 | 6,465 | 46.9 | 8,713 | 94.2 | 1,956 | 59.1 | 3,118 | **72.50**$_{+0.98}$ | **4,191**$_{-42.6\%}$ |
| *Qwen3-8B* | | | | | | | | | | | | |
| Vanilla | 95.4 | 2,370 | 75.1 | 15,425 | 65.2 | 18,401 | 93.4 | 5,577 | 55.6 | 8,741 | 76.94 | 10,103 |
| GRPO | 95.8 | 2,355 | 74.0 | 15,061 | 65.4 | 17,987 | 94.4 | 5,440 | 55.8 | 8,819 | 77.08$_{+0.14}$ | 9,932$_{-1.7\%}$ |
| RL + Length Penalty | 95.4 | 1,323 | 73.8 | 9,666 | 60.0 | 11,547 | 94.2 | 3,247 | 56.2 | 5,293 | 75.92$_{-1.02}$ | 6,215$_{-38.5\%}$ |
| Short-RL | 95.7 | 1,241 | 73.0 | 9,972 | 60.9 | 10,937 | 93.2 | 3,048 | 55.9 | 4,987 | 75.74$_{-1.20}$ | 6,037$_{-40.2\%}$ |
| S-GRPO | 96.1 | 1,292 | 77.3 | 8,810 | – | – | 95.2 | 3,166 | 57.7 | 5,271 | – | – |
| ThoughtFold | 96.2 | 1,097 | 78.1 | 9,099 | 65.4 | 11,670 | 97.4 | 2,933 | 57.9 | 4,571 | **79.00**$_{+2.06}$ | **5,874**$_{-41.9\%}$ |
| *Qwen3-14B* | | | | | | | | | | | | |
| Vanilla | 95.5 | 1,909 | 75.4 | 14,116 | 69.2 | 16,978 | 95.2 | 5,078 | 58.8 | 7,576 | 78.82 | 9,131 |
| GRPO | 96.1 | 1,956 | 77.7 | 14,544 | 69.7 | 16,689 | 95.8 | 5,140 | 59.3 | 7,966 | 79.72$_{+0.90}$ | 9,259$_{+1.4\%}$ |
| RL + Length Penalty | 95.8 | 1,090 | 74.8 | 9,056 | 61.5 | 11,376 | 95.8 | 2,866 | 59.4 | 4,949 | 77.46$_{-1.36}$ | 5,867$_{-35.7\%}$ |
| Short-RL | 96.1 | 1,034 | 75.2 | 9,255 | 63.1 | 11,392 | 96.4 | 2,789 | 59.8 | 4,685 | 78.12$_{-0.70}$ | 5,831$_{-36.1\%}$ |
| S-GRPO | 96.3 | 952 | 77.9 | 8,932 | – | – | 96.4 | 2,652 | 60.6 | 4,537 | – | – |
| ThoughtFold | 96.8 | 894 | 79.3 | 8,869 | 69.7 | 11,217 | 97.1 | 2,577 | 60.9 | 4,121 | **80.76**$_{+1.94}$ | **5,536**$_{-39.4\%}$ |

DEER (Yang et al., 2025) and off-policy compression techniques such as ConCISE (Qiao et al., 2025). Due to code unavailability, we report the official results for S-GRPO.

**Training Details.** We use the DeepMath-103K dataset (He et al., 2025) for training. This large-scale dataset features challenging mathematics problems spanning multiple difficulty levels (ranging from grade 5 to grade 10), providing necessary complexity for effective reasoning optimization. For both ThoughtFold and GRPO, we set the generation and training batch sizes to $128 \times 8$, with a maximum response length of 30k tokens. ThoughtFold uses a learning rate of $1 \times 10^{-6}$ and a trade-off coefficient of $\lambda = 0.1$. For the Short-RL baseline, we set $\alpha = 1$ following its original implementation (Yuan et al.). Regarding the attention computation in ThoughtFold, we utilize the attention map from the middle Transformer layer. Notably, the self-reflection process requires only a single computation per correct sample, incurring negligible training overhead.

### 4.2. Experimental Results

**Main Results.** Table 1 demonstrates that ThoughtFold outperforms existing RLVR baselines across four reasoning models and five benchmarks. Compared to vanilla models, our method improves absolute accuracy by 0.98%–2.82% while compressing sequence length by 39.4%–56.1%. Against standard GRPO, ThoughtFold not only secures an accuracy advantage of 1.04%–2.24% but also achieves substantial length reductions of 40.2%–56.2%. Crucially, it surpasses the state-of-the-art efficient reasoning method,

S-GRPO, in both accuracy and efficiency metrics. Beyond gains on in-domain math benchmarks (e.g., GSM8K, AIME), ThoughtFold exhibits superior generalization on out-of-domain scientific tasks (e.g., GPQA). This suggests that our fine-grained preference learning fosters robust, transferable reasoning structures rather than mere solution memorization. Furthermore, ThoughtFold demonstrates adaptive reasoning: it aggressively shortens chains for simpler tasks like GSM8K, while for complex benchmarks (e.g., AIME 2024/2025), it maintains efficiency comparable to length-regularized methods without accuracy degradation.

**Ablation Study.** To validate the individual contributions of each component within ThoughtFold, we conducted ablation studies on Qwen3-8B under three specific configurations: (1) *w/o attention*: substitutes the attention-based importance metric with a random selection policy during the Internal Folding phase; (2) *w/o Internal Folding*: omits the fine-grained pruning phase (Internal Folding), relying solely on coarse truncation (Tail Truncation); (3) *w/o mask*: disables the dynamic masking strategy, reverting to global trajectory-level preference signals.

First, the full ThoughtFold configuration obtains the best performance on both accuracy (79.00%) and efficiency (5,874 tokens), consistently outperforming both the random policy and the coarse-only approach. This superiority is particularly evident on complex reasoning benchmarks, confirming that our attention-guided pruning effectively distinguishes essential logic from off-target attempts, enabling the model

*Table 2.* Ablation study on Qwen3-8B.

| Method | Phase 1 | Phase 2 | Mask | GSM8K | | AIME 2024 | | AIME 2025 | | MATH-500 | | GPQA | | Overall | |
|---|---|---|---|---|---|---|---|---|---|---|---|---|---|---|---|
| | | | | Acc | Tokens | Acc | Tokens | Acc | Tokens | Acc | Tokens | Acc | Tokens | Acc | Tokens |
| *Qwen3-8B* | | | | | | | | | | | | | | | |
| GRPO | N | N | N | 95.8 | 2,355 | 74.0 | 15,061 | 65.4 | 17,987 | 94.4 | 5,440 | 55.8 | 8,819 | 77.08 | 9,932 |
| **Ours** | Y | Y | Y | 96.2 | 1,097 | 78.1 | 9,099 | 65.4 | 11,670 | 97.4 | 2,933 | 57.9 | 4,571 | **79.00**+1.92 | **5,874**-40.9% |
| *w/o attention* | random | Y | Y | 95.5 | 1,398 | 77.1 | 9,978 | 63.9 | 13,114 | 96.1 | 3,194 | 56.9 | 4,873 | 77.90+0.82 | 6,511-34.4% |
| *w/o Internal Folding* | N | Y | Y | 96.3 | 1,457 | 77.9 | 10,449 | 64.6 | 13,081 | 96.9 | 3,367 | 57.7 | 4,786 | 78.68+1.60 | 6,628-33.3% |
| *w/o mask* | Y | Y | N | 94.8 | 1,134 | 73.2 | 11,079 | 62.3 | 13,371 | 94.1 | 3,542 | 54.6 | 5,099 | 75.80-1.28 | 6,845-31.1% |

to condense reasoning paths without compromising semantic depth. Second, the *w/o mask* setting results in severe performance degradation, with overall accuracy dropping to 75.80%—falling even below the standard GRPO baseline (77.08%). We attribute this failure to *credit assignment ambiguity*: without step-level masking, identical reasoning steps appearing in both preferred (concise) and rejected (redundant) trajectories receive conflicting gradient updates. This confirms that our dynamic masking strategy is essential for establishing precise fine-grained signals, thereby guaranteeing both reasoning accuracy and efficiency.

**Hyperparameter Analysis.** We examine the effect of the coefficient $\lambda$, which regulates the trade-off between GRPO and fine-grained preference learning signal. Table 3 demonstrates that ThoughtFold achieves a stable and controllable trade-off between reasoning accuracy and efficiency.

*Table 3.* Hyperparameter analysis of the trade-off coefficient $\lambda$.

| Coefficient $\lambda$ | 0 (GRPO) | 0.001 | 0.01 | 0.1 | 1.0 | $\infty$ (MDPO) |
|---|---|---|---|---|---|---|
| **Accuracy (%)** ↑ | 77.08 | 78.14 | 79.21 | 79.00 | 77.23 | 73.57 |
| **Tokens** ↓ | 9,932 | 7,512 | 6,431 | 5,874 | 5,249 | 4,787 |

### 4.3. Analysis

**Minimum Average Length@k** To further investigate reasoning efficiency of the models, we introduce the *Minimum Average Length@k* (ML@$k$) metric. Analogous to pass@k, ML@$k$ estimates the expected minimum sequence length given a budget of $k$ rollouts. For $n$ independent rollout lengths sorted as $l_1 \leq l_2 \leq \cdots \leq l_n$, the metric is formulated as:

$$\text{ML@}k = \sum_{i=1}^{n-k+1} l_i \times \frac{\binom{n-i}{k-1}}{\binom{n}{k}} \quad (11)$$

where $\binom{n}{k}$ is the binomial coefficient. The detailed derivation is provided in the Appendix B. Figure 3 illustrates the ML@$k$ curves ($k = 1 \ldots 32$) of Qwen3-8B, length-reward baseline Short-rl and ThoughtFold on AIME benchmarks. ThoughtFold demonstrates substantial efficiency gains over Qwen3-8B. Notably, while Short-RL and ThoughtFold exhibit comparable average lengths (ML@1), ThoughtFold displays a significantly steeper decay and lower bound as $k$ increases. This suggests that ThoughtFold does not merely reshape the global length distribution. Instead, it optimizes the underlying reasoning structure for concise reasoning.

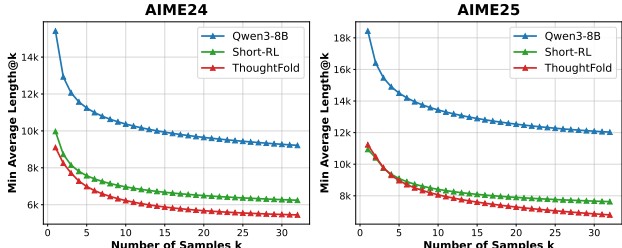

*Figure 3.* Comparison of expected minimum reasoning length (ML@$k$) on AIME. ThoughtFold exhibits a sharper decay and achieves a significantly lower minimum length as k increases.

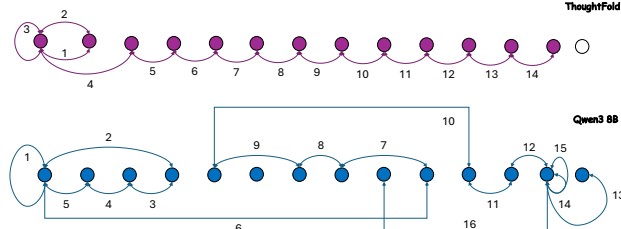

*Figure 4.* Visualization of reasoning topology via concept graphs. ThoughtFold (top) exhibits a linear structure by eliminating the redundant loops and backtracking found in the baseline (bottom).

**Visualizing Reasoning Topology.** To intuitively understand the impact of ThoughtFold on reasoning structure, we visualize the temporal progression of traces using a concept graph adapted from Minegishi et al. (2025). Reasoning steps are mapped to discrete concept nodes via clustering, where nodes at the same horizontal position represent identical semantic concepts. As shown in Figure 4, vanilla Qwen3-8B exhibits a chaotic topology, characterized by long-range backtracking jumps (e.g., edge 6) and local repetitive loops (e.g., edges 14-15). These patterns visualize the redundancy and hesitation in vanilla generation. Conversely, ThoughtFold presents a highly linear and sequential structure. By effectively folding redundant explorations, our method produces a concise trajectory that moves directly from problem to solution without semantic detours. This structural efficiency aligns with our geometric analysis (Appendix A), which confirms that ThoughtFold compresses heavy-tailed outliers into a stable, compact distribution.

**Case study.** Figure 5 presents a representative case comparing the CoT reasoning of Qwen3-8B, Short-RL, and ThoughtFold. While Qwen3-8B arrives at the correct answer, its reasoning path is verbose, characterized by redun-

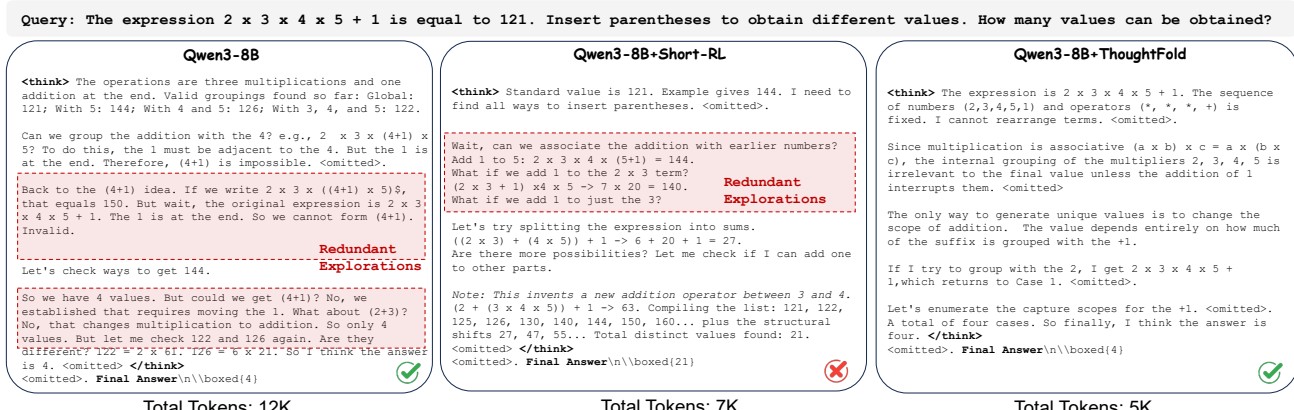

*Figure 5.* Comparison of a generated content sample on GSM8K. The vanilla model gets the correct answer but repeats itself unnecessarily. Short-RL is misled by its own redundant explorations. Despite reasoning fast, it drifts into hallucinations and results in a wrong answer. In contrast, ThoughtFold produces a concise and correct chain of thought.

dant revisions of previously established points. Conversely, although Short-RL successfully compresses the reasoning process, it leads to hallucinations and wrong answers. This stems from the outcome-based length reward, which indiscriminately incentivizes incorrect reasoning steps, thereby degrading accuracy in the pursuit of brevity. ThoughtFold achieves accurate and efficient reasoning. It demonstrates superior intuition by directly deriving key insights and bypasses the preliminary example construction observed in other methods, resulting in a concise solution path.

## 5. Related Works

**RLVR.** The paradigm of training Large Reasoning Models (LRMs) has shifted from supervised fine-tuning to Reinforcement Learning with Verifiable Rewards (RLVR) (Xu et al., 2025a). Leveraging ground-truth labels as sparse reward signals, RLVR encourages models to explore complex reasoning paths and self-correcting behaviors (DeepSeek-AI et al., 2025; Stechly et al., 2025; Comanici et al., 2025). Prominent algorithms, most notably GRPO (DeepSeek-AI et al., 2025), achieve training stability by normalizing rewards across group samples to estimate advantages. However, a fundamental limitation of standard RLVR is the coarse credit assignment problem. Since the reward is determined solely by the final outcome, every step in a correct trajectory—whether it is a crucial deduction or a redundant loop—receives identical reinforcement. This indiscriminate positive signal inevitably leads to the "overthinking" phenomenon (Sui et al., 2025; Chen et al., 2025). Our work builds upon the RLVR foundation but addresses this inherent inefficiency by introducing fine-grained preference learning.

**Efficient Reasoning.** Existing approaches for efficient reasoning generally fall into two categories: training-free and training-based methods. Training-free methods typically rely on dynamic prompting strategies (Han et al., 2024; Xu

et al., 2025b; Lee et al., 2025; Renze & Guven, 2024), adaptive sampling pruning (Xie et al., 2023; Liao et al., 2025; Li et al., 2024), or inference-time early-exit mechanisms (Ma et al., 2025a; Yang et al., 2025) to reduce computation without updating model parameters. Training-based methods seek to internalize efficiency into the model itself. Initial efforts focused on Supervised Fine-Tuning (SFT) with concise CoT data (Yu et al., 2024; Kang et al., 2025; Xia et al., 2025; Ma et al., 2025b). Recently, Reinforcement Learning (RL) has emerged as a dominant paradigm, where models are trained with length-regularized rewards (Team et al., 2025b; Luo et al., 2025; Aggarwal & Welleck, 2025; Arora & Zanette, 2025a; Yeo et al., 2025; Gui et al., 2026b). A representative recent work is S-GRPO (Dai et al., 2025), which shifts from parallel sampling to sequential early-exit sampling, incentivizing the model to terminate reasoning as soon as the correct answer is reachable. However, both standard length-penalty methods and S-GRPO operate primarily on outcome-based supervision. They lack the granularity to penalize specific redundant steps within a correct trajectory, leading to the potential memorization of useless explorations. While step-level value estimation (Yue et al., 2025) offers precise signals, its reliance on Monte Carlo rollouts incurs prohibitive computational costs, rendering it unscalable for training large models. In contrast, Thought-Fold introduces an efficient, fine-grained preference learning framework that leverages an introspective mechanism to identify and explicitly penalize internal redundancies rather than merely rewarding shorter outcomes, effectively folding reasoning chains into concise paths.

## 6. Conclusion

We introduce ThoughtFold, a framework for efficient reasoning that integrates an introspective strategy with fine-grained preference learning to explicitly penalize redundancy. Experiments demonstrate that ThoughtFold achieves superior

efficiency-accuracy trade-offs with strong generalization. Our analysis indicates that ThoughtFold fundamentally reshapes the reasoning topology, creating direct solution paths rather than simply truncating verbose ones. Mechanistically, it acts as a stabilizing operator, yielding more compact trajectory representations in the embedding space.

## Impact Statement

This paper presents work whose goal is to advance the field of machine learning. There are many potential societal consequences of our work, none of which we feel must be specifically highlighted here.

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

## A. Quantile radius analysis

We characterize the resulting representation distributions using the Effective Radius. First, we calculate centroids for individual reasoning steps and derive a global mean $\mu$ to serve as the reference refin. We compute Principal Components jointly to ensure common scaling. Crucially, to ensure we are measuring true statistical variation rather than just the magnitude of principal components, we employ Mahalanobis distance instead of standard Euclidean distance.For a set of step centroids $S$, the effective radius $R_q$ is defined as the minimum radius required for a Mahalanobis ball centered at $\mu$ to capture $q\%$ of the probability mass:

$$R_q(S) = \inf \left\{ r \ : \ \frac{1}{|S|} \sum_{z \in S} \mathbb{I}\left(d_M(z, \mu) \leq r\right) \geq q \right\}.$$

Figure 6 presents the Radius Difference $\Delta R_q = R_q(\text{Base}) - R_q(\text{Model})$. The dashed grey line at $y = 0$ represents the Base model's geometry; values above this line indicate the model is more compressed than the Base, while values below indicate it is more dispersed.

ThoughtFold (Left): A Dual Geometric Regularizer. The ThoughtFold curve reveals a sophisticated "dip-then-spike" relationship relative to the Base model. *Core Expansion ($q \in [20, 75]$):* In the distributional core, $\Delta R$ is consistently negative. This implies that ThoughtFold's radius is slightly larger than the Base model's. By expanding the volume of the core, ThoughtFold maintains a more "profuse" solution space, preserving semantic diversity and preventing mode collapse during reasoning. *Tail Compression ($q > 80$):* Conversely, at high quantiles, we observe a sharp positive spike ($\Delta R > 0$). This indicates a significant reduction in radius compared to the Base model. ThoughtFold effectively identifies and "folds" the Base model's heavy-tailed outliers back into a compact.

Short-RL (Right): Tail Explosion and Instability. The Short-RL model (right panel) exhibits a fundamentally different geometric profile. *Core Similarity:* For the vast majority of the distribution ($q < 95$), the difference is negligible ($\Delta R \approx 0$), suggesting that standard RL fine-tuning leaves the semantic core of the Base model largely structurally unchanged. *Extreme Divergence ($q \approx 100$):* At the extreme tail, the curve plummets to a massive negative value ($\Delta R \approx -4$). Since $\Delta R = R_{\text{Base}} - R_{\text{Short-RL}}$, this implies $R_{\text{Short-RL}}$ is exponentially larger than $R_{\text{Base}}$. Instead of regularizing the solution space, Short-RL exacerbates the heavy-tailed nature of the distribution, pushing outliers even further away from the centroid. This "tail explosion" suggests that without the geometric constraints introduced by ThoughtFold, RL-driven optimization can destabilize the model at the fringes, leading to erratic exploration far from the semantic manifold of valid reasoning.

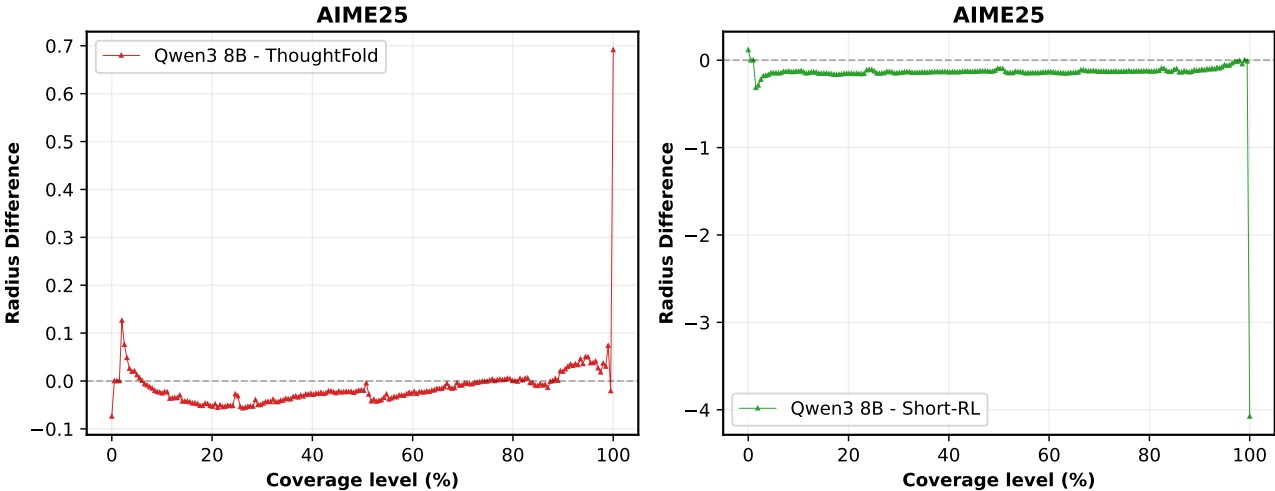

*Figure 6.* **Quantile Radius Difference** ($\Delta R = R_{\text{Base}} - R_{\text{ThoughtFold}}$). The plot reveals a dual geometric phenomenon. In the representational core (quantiles 20–60), the difference is negligible or slightly negative, indicating that ThoughtFold preserves semantic diversity. In the extreme tail (quantiles > 80), we observe a sharp positive spike, indicating that ThoughtFold significantly compresses the heavy-tailed outliers of the base model.

## B. Derivation of expected minimum length

We present a detailed derivation for equation (11). The derivation is based on similar procedure as Pass @$k$ in Chen et al. (2021). Let $\mathcal{L} = \{l_1, l_2, \ldots, l_n\}$ denote the set of average lengths obtained from $n$ independent rollouts. We assume the set is sorted in ascending order such that $l_1 \leq l_2 \leq \cdots \leq l_n$. The goal is to derive the expected value of the minimum element within a subset of size $k$ ($k \leq n$) drawn uniformly at random from $\mathcal{L}$ without replacement. Let $S \subset \mathcal{L}$ be such a random subset where $|S| = k$, and let random variable $X = \min(S)$. By the definition of expectation for a discrete random variable, the expected minimum length is given by the summation of each element $l_i$ multiplied by the probability that $l_i$ is the minimum of the subset $S$:

$$\text{ML@}k = \mathbb{E}[X] = \sum_{i=1}^{n} l_i \cdot P(X = l_i)$$

To determine the probability $P(X = l_i)$, we consider the total number of possible subsets and the number of favorable subsets where $l_i$ is the minimum.Total Subsets: The total number of ways to choose a subset of size $k$ from a set of size $n$ is given by the binomial coefficient:

$$N_{\text{total}} = \binom{n}{k}$$

For a specific length $l_i$ to be the minimum of the subset $S$, two conditions must be satisfied: The element $l_i$ must be included in $S$. The remaining $k-1$ elements of $S$ must be chosen strictly from elements greater than or equal to $l_i$.Since $\mathcal{L}$ is sorted, the elements greater than or equal to $l_i$ (excluding $l_i$ itself for the distinct selection) correspond to the indices $\{i+1, i+2, \ldots, n\}$. The count of such eligible elements is $n-i$.Therefore, the number of ways to complete the subset is equivalent to choosing $k-1$ elements from the $n-i$ available elements:

$$N_{\text{favorable}}(l_i) = 1 \times \binom{n-i}{k-1}$$

The probability that $l_i$ is the minimum is the ratio of favorable outcomes to total outcomes:

$$P(X = l_i) = \frac{\binom{n-i}{k-1}}{\binom{n}{k}}$$

We note that $\binom{n-i}{k-1} = 0$ if $n - i < k - 1$. Therefore, the index $i$ must satisfy $n - i \geq k - 1$, which implies $i \leq n - k + 1$. This imposes an upper bound on the summation, as $l_i$ cannot be the minimum if there are insufficient elements larger than it to fill the subset.ConclusionSubstituting the probability term and the summation limit back into the expectation formula, we arrive at the estimator:

$$\text{ML@}k = \sum_{i=1}^{n-k+1} l_i \times \frac{\binom{n-i}{k-1}}{\binom{n}{k}},$$

as desired.

## C. Implementation Details

### C.1. Algorithm Details

We present the algorithm for constructing preference pairs (Alg. 1) and the training process of ThoughtFold (Alg. 2).

### C.2. Prune-and-Verify Protocol

For each pruned candidate, ThoughtFold verifies whether the model can still derive the correct final answer from the shortened reasoning context. To avoid noisy decisions from stochastic generation, we use $K = 4$ parallel verification rollouts rather than a single rollout. Let $y^{(k)}$ be the $k$-th continuation and $a^\star$ be the ground-truth answer. We compute

$$\hat{p}_{\text{verify}} = \frac{1}{4} \sum_{k=1}^{4} \mathbb{I}\left[ \text{Verify}(y^{(k)}, a^\star) = 1 \right]. \tag{12}$$

A pruned candidate is accepted only if $\hat{p}_{\text{verify}} \geq 0.75$, i.e., at least three out of four continuations are correct. Otherwise, it is rejected and not used as a reliable positive folding target. During verification, continuations with abnormal formats, such as repeated thinking tags or excessively long answers, are also marked as invalid. This protocol reduces the risk of constructing noisy preference pairs from single-sample verification.

---

**Algorithm 1** Introspective Redundancy Identification

---

1: **Input:** Query $x$, Correct Trajectory $\tau_{ref} = (z_{ref}, y^*)$, Model $\pi_\theta$, Max Iterations $N_{max,1}, N_{max,2}$
2: **Output:** Preference Dataset $\mathcal{D}_{pair}$
3: $\mathcal{D}_{pair} \leftarrow \emptyset; \quad z_{best} \leftarrow z_{ref}$
4:                                                        ▷ Phase 1: Tail Truncation (Binary Search on Length)
5: $L \leftarrow 0, R \leftarrow |z_{ref}|, i \leftarrow 0$
6: **while** $L < R$ **and** $i < N_{max,1}$ **do**
7:    $m \leftarrow \lfloor (L+R)/2 \rfloor; \quad z_{cand,i} \leftarrow z_{ref}[:m]$
8:    $\hat{y}_{cand,i} \sim \pi_\theta(\cdot | x, z_{cand,i})$
9:    **if** $\hat{y}_{cand,i} == y^*$ **then**
10:       $M_{w,l} \leftarrow \text{DynamicMaskStrategy}(z_{cand,i}, z_{best})$                       ▷ Sec. 3.2
11:       $\mathcal{D}_{pair}.\text{append}((z_{cand,i}, z_{best}, M_{w,l}))$                       ▷ Concise Success
12:       $z_{best} \leftarrow z_{cand,i}; \quad R \leftarrow m$
13:    **else**
14:       $M_{w,l} \leftarrow \text{DynamicMaskStrategy}(z_{best}, z_{cand,i})$
15:       $\mathcal{D}_{pair}.\text{append}((z_{best}, z_{cand,i}, M_{w,l}))$                       ▷ Over-Simplified Failure
16:       $L \leftarrow m + 1$
17:    **end if**
18:    $i \leftarrow i + 1$
19: **end while**
20:                                      ▷ Phase 2: Internal Folding (Binary Search on Retention Ratio)
21: Compute importance $I(s_t)$ for $s_t \in z_{best}$                                 ▷ Eq. (3)
22: $k_{min} \leftarrow 0, k_{max} \leftarrow 1.0, j \leftarrow 0$
23: **while** $k_{max} - k_{min} > \epsilon$ **and** $j < N_{max,2}$ **do**
24:    $k \leftarrow (k_{min} + k_{max})/2$
25:    $N_{keep} \leftarrow \lceil k \cdot |z_{best}| \rceil$
26:    $z_{cand,j} \leftarrow \{ s_t \in z_{best} \mid \text{rank}(I(s_t)) \leq N_{keep} \}$                 ▷ Keep top-k steps based on importance score
27:    $\hat{y}_{cand,j} \sim \pi_\theta(\cdot | x, z_{cand,j})$
28:    **if** $\hat{y}_{cand,j} == y^*$ **then**
29:       $M_{w,l} \leftarrow \text{DynamicMaskStrategy}(z_{cand,j}, z_{best})$
30:       $\mathcal{D}_{pair}.\text{append}((z_{cand,j}, z_{best}, M_{w,l}))$                     ▷ Concise Success
31:       $z_{best} \leftarrow z_{cand,j}; \quad k_{max} \leftarrow k$
32:    **else**
33:       $M_{w,l} \leftarrow \text{DynamicMaskStrategy}(z_{best}, z_{cand,j})$
34:       $\mathcal{D}_{pair}.\text{append}((z_{best}, z_{cand,j}, M_{w,l}))$                     ▷ Over-Simplified Failure
35:       $k_{min} \leftarrow k$
36:    **end if**
37:    $j \leftarrow j + 1$
38: **end while**
39: **return** $\mathcal{D}_{pair}$

---

---

**Algorithm 2** Joint Group Relative and Fine-Grained Reasoning Policy Optimization

---

**Require:** Dataset $\mathcal{D}$, Policy $\pi_\theta$, Reference Model $\pi_{\text{ref}}$, Group Size $G$, Coefficient $\lambda$
1: Initialize policy parameters $\theta \leftarrow \theta_{\text{init}}$
2: **while** not converged **do**
3:                                                                       ▷ Step 1: Sampling and Evaluation
4:     Sample batch $\mathcal{B} = \{x_1, \ldots, x_B\} \sim \mathcal{D}$
5:     Generate group $\{\tau_i\}_{i=1}^G \sim \pi_\theta(\cdot|x)$ for each $x \in \mathcal{B}$
6:     Compute rewards $r(y_i)$ and Group Advantages $A_i$         ▷ GRPO Eq. (3)
7:                             ▷ Step 2: Data Construction
8:     $\mathcal{D}_{pair} \leftarrow \emptyset$
9:     **for** each query $x$ in $\mathcal{B}$ **do**
10:       $\mathcal{T}_{\text{correct}} \leftarrow \{\tau_i \mid r(y_i) = 1\}$
11:       **for** $\tau \in \mathcal{T}_{\text{correct}}$ **do**
12:         $\mathcal{D}_{pair} \leftarrow \mathcal{D}_{pair} \cup \text{SELFREFLECTIVEPREFERENCECONSTRUCTION}(x, \tau)$     ▷ Alg. 1
13:       **end for**
14:     **end for**
15:                            ▷ Step 3: Joint Policy Update
16:     **for** iteration $k = 1, \ldots, \mu$ **do**
17:       Compute $\mathcal{J}_{\text{GRPO}}(\theta)$ using advantages $A$ and importance ratio $\rho$
18:       Compute $\mathcal{J}_{\text{MDPO}}(\theta)$ on $\mathcal{D}_{pair}$ using masks $M$     ▷ Eq. (9)
19:       $\mathcal{J}_{\text{total}} \leftarrow \mathcal{J}_{\text{GRPO}}(\theta) + \lambda \mathcal{J}_{\text{MDPO}}(\theta)$
20:       Update $\theta$ by maximizing objective: $\theta \leftarrow \theta + \eta \nabla_\theta \mathcal{J}_{\text{total}}$
21:     **end for**
22: **end while**
**Ensure:** Optimized policy $\pi_\theta$

---

# D. More Related Works

## D.1. Process Reward Models

Process reward models (PRMs) provide step-level supervision for reasoning trajectories, typically by identifying incorrect or suboptimal intermediate steps. For example, OmegaPRM (Luo et al., 2024) constructs process-supervision data for mathematical reasoning by using Monte Carlo tree search and binary search to locate the first erroneous step. Recent works further improve PRM data construction or stepwise correction for mathematical reasoning (Sun et al., 2025; Wu et al., 2025).

Although these methods also analyze intermediate reasoning steps, their objective differs from ours. PRM-based methods usually assign rewards or labels to individual steps for training a verifier or reward model. In contrast, ThoughtFold targets efficient reasoning: it identifies redundant segments in outcome-correct reasoning chains and directly optimizes the policy to bypass them. Thus, our supervision is not a step-level reward, but a preference over shortcut transitions between reasoning steps.

Binary search also plays a different role. In OmegaPRM, it is used to locate first errors for PRM data construction; in ThoughtFold, it is only an efficient search strategy in prune-and-verify for finding foldable redundancy. The core of ThoughtFold is introspective redundancy identification and fold-anchor-based dynamic masking, which enables precise preference learning over step transitions while avoiding conflicting supervision on shared correct prefixes.

## D.2. Reinforcement Learning

Reinforcement learning (RL) provides a general framework for optimizing sequential decision-making policies through reward feedback (Sutton & Barto, 2018). With deep neural networks, RL has achieved strong results in complex control and planning problems, such as value-based decision making (Mnih et al., 2015), tree-search-based game playing (Silver et al., 2016), and stable policy optimization with PPO (Schulman et al., 2017). It has also become a key optimization paradigm for aligning and improving large language models, especially through preference-based objectives such as RLHF (Christiano et al., 2017; Ouyang et al., 2022). Beyond these applications, RL has also been applied to practical optimization scenarios, such as chip design (Geng et al., 2024; 2025; Wang et al., 2026), as well as to improving the faithfulness of large-model reasoning (Gui et al., 2026a).

# E. Justification of Attention-based Pruning

In the internal folding stage, ThoughtFold ranks reasoning steps according to their contribution to answer generation. This section provides additional justification for using attention scores as the step-importance proxy and for adopting middle-layer attention by default.

**Why attention can indicate reasoning-step importance.** The intuition is that the model itself has implicit awareness of which reasoning steps are useful for producing the final answer. Compared with the final answer, the reasoning chain is usually much longer and often contains redundant exploration, repeated verification, or unnecessary detours. Therefore, a natural way to estimate the importance of a reasoning step is to measure how much the answer-generation tokens attend to that step.

This choice is also supported by prior work on efficient LLM inference (Zhang et al., 2023; Luo et al., 2026; Liu et al., 2025; Chen et al., 2026). H2O (Zhang et al., 2023) shows that attention scores can serve as an effective proxy for token importance when identifying heavy-hitter tokens in generative inference. More directly related to long reasoning models, FROST (Luo et al., 2026) observes that attention over reasoning steps during answer generation is highly sparse: only a subset of reasoning steps receives most of the attention, while many steps receive little attention. These findings support the use of answer-to-reasoning attention as a practical proxy for identifying low-contribution reasoning steps.

Formally, for a reasoning step $s_i$ and the final answer tokens $\mathcal{A}$, we compute its importance score by averaging the attention from answer tokens to tokens in $s_i$ at a selected transformer layer:

$$I(s_i) = \frac{1}{|\mathcal{A}|} \sum_{a \in \mathcal{A}} \frac{1}{|s_i|} \sum_{t \in s_i} \mathrm{Attn}^{(\ell)}(a, t), \tag{13}$$

where $\ell$ denotes the selected attention layer. Steps with lower importance scores are considered less directly involved in answer generation and are therefore prioritized for pruning and verification. Importantly, the attention score is not used as a standalone deletion rule. A step is treated as foldable only when the pruned trajectory can still lead to a correct answer under the prune-and-verify procedure.

**Why middle-layer attention is used.** By default, we use the middle transformer layer for importance computation. This choice follows the observation that shallow-layer representations are often relatively noisy and dominated by local lexical patterns, while intermediate layers tend to encode more informative and transferable representations. Recent studies on language-model representations also show that intermediate layers can provide stronger embeddings and more useful hidden representations than either very shallow or final layers (Skean et al., 2025; Skean et al.). Thus, middle-layer attention provides a reasonable balance: it is less noisy than early-layer attention and less overly specialized to final-token prediction than very deep-layer attention.

**Sensitivity to layer choice.** To verify that ThoughtFold does not rely on a fragile layer-selection heuristic, we conduct a sensitivity analysis on Qwen3-8B. We compare four choices for computing the attention-based importance score: first layer, middle layer, last layer, and the average over all layers. As shown in Table 4, the middle layer achieves the best accuracy and shortest average response length, while the other choices only lead to mild degradation. This suggests that the attention-based pruning criterion is relatively robust to the exact layer choice.

*Table 4.* Sensitivity analysis of attention layer choice on Qwen3-8B. The middle layer performs best, while other choices degrade only mildly.

| Layer | First | Middle | Last |
|---|---|---|---|
| Accuracy | 78.72 | **79.00** | 78.85 |
| Tokens | 5899 | **5874** | 5911 |

Overall, attention is used in ThoughtFold as a practical and empirically validated proxy for reasoning-step importance. We do not assume that attention perfectly explains all reasoning dependencies. Instead, attention provides an efficient candidate-ranking signal, while the prune-and-verify mechanism determines whether a low-attention step can actually be folded without harming correctness.

