# OpenReview forum: "ThoughtFold: Folding Reasoning Chains via Introspective Preference Learning"
_ICML.cc/2026/Conference — ICML 2026 regular_

### Official Review · Reviewer_zCHd · 2026-03-13

**Soundness:** 3
**Presentation:** 3
**Significance:** 3
**Originality:** 3
**Overall Recommendation:** 4
**Confidence:** 4

**Summary:**

This paper tackles the overthinking problem in large reasoning models trained with RLVR. The core observation is that outcome-based rewards indiscriminately reinforce both useful deductions and redundant explorations within correct trajectories. The authors propose ThoughtFold, which uses an introspective two-phase strategy to identify redundant reasoning steps: first tail truncation via binary search on prefix length, then internal folding guided by attention-based importance scores. The identified redundancies are used to construct fine-grained preference pairs, optimized with a masked DPO objective jointly with GRPO. Experiments on GSM8K, MATH-500, AIME 2024/2025, and GPQA across four models show 39 to 56 percent token reduction while maintaining or improving accuracy over baselines.

**Compliance With Llm Reviewing Policy:**

Affirmed.

**Final Justification:**

My final recommendation is to keep my score, the rebuttal has addressed my concerns but by reevaluating the paper, I may give up the opportunity to give it a 5 which explicitly indicate I love this work very much, I think this is good, but not extraordinary to me. Giving a positive but not that strong recommendation would be suitable for me.

**Key Questions For Authors:**

1. How sensitive is the method to the choice of transformer layer for the attention-based importance computation? Have you tried averaging across layers or using the last layer instead? This matters because attention distributions vary significantly across depth.

2. The introspective identification requires generating answers from pruned trajectories during training. How many additional forward passes does this add per training step in practice, and how does wall-clock training time compare to standard GRPO?

3. For the internal folding phase, steps are ranked by importance and removed in order of lowest importance. But removing non-contiguous steps creates artificial token sequences the model has never seen during pretraining. Did you observe any issues with the model producing correct answers from these Frankenstein sequences simply by pattern matching the remaining fragments?

4. The hyperparameter analysis in Table 3 shows that lambda=0.01 actually gives the best accuracy, yet the paper uses lambda=0.1. What motivated this choice, and is the selected operating point consistent across all four base models?

If the author can address my concern, I would be willing to increase my score.

**Limitations:**

To me,  no.  And it would benefit from discussing the reliance on math-only training data and whether the approach works when training on mixed-domain datasets I think.

**Strengths And Weaknesses:**

Strengths:
- The introspective redundancy identification via prune-and-verify is a natural and principled way to obtain step-level supervision without expensive Monte Carlo rollouts.
- The dynamic masking strategy for DPO is well designed, addressing the credit assignment conflict that arises when the same steps appear in both preferred and rejected trajectories.
- Comprehensive evaluation across four models of two families and five benchmarks of varying difficulty, with consistent gains in both accuracy and efficiency.
- The analysis section is above average: ML@k curves, concept graph visualization, and the quantile radius analysis together provide converging evidence that the method changes reasoning structure rather than just shortening outputs.

Weaknesses:
- The attention-based importance score uses only a single middle layer, with no justification or sensitivity analysis for this choice across different model architectures.
- S-GRPO results are missing for AIME 2025 across all four models, making the primary advanced baseline comparison incomplete on the hardest benchmark.
- All training uses math-only data, yet the paper claims generalization to GPQA without discussing whether the efficiency gains come from learned reasoning patterns or domain-specific shortcuts that happen to transfer.

---

> ### Author Rebuttal · Authors · 2026-03-29
>
> Thank you for your constructive comment. Following are our responses to each comment:
> ## Response to Weakness1 and Question1 about attention-based importance:
> > The attention-based importance score uses a single middle layer, with no justification or sensitivity analysis.
> - **Why attention?** Due to limited space, please refer to our response to R-Ttnk (W2: Soundness) and R-NRrv for the detailed justification.
> - **Why middle-layer attention?** Prior studies (e.g., [1,2]) suggest that shallow-layer representations are relatively noisy, while intermediate layers are more informative than deeper layers and provide better embeddings. We therefore use middle-layer attention.
> - **Sensitivity analysis.** We conduct sensitivity analysis of layer choice on Qwen3-8B. The result shows that  the middle layer performs best, with other choices degrading only mildly. This confirms ThoughtFold's robustness to layer selection and validates attention as a stable importance proxy. The results are reported below for reference:
> |Layer|First|Middle|Last|All-Layer-Avg|
> |-|-|-|-|:-:|
> |ACC|78.72|79.01|78.85|78.91|
> |Tokens|5899|5874|5911|5893|
>
> We will include the above discussion in the next version.
> ## Response to Weakness2:
> > S-GRPO results are missing for AIME 2025, making the comparison incomplete on the hardest benchmark.
>
> S-GRPO does not release its code, so we cite its results for fairness. Note that AIME24 is already a hard benchmark that provides a rigorous comparison. We have actually reproduced S-GRPO ourselves before, and we report its AIME25 results below for reference:
> |ACC/Tokens|R1-distill-7B|R1-distill-14B|Qwen3-8B|Qwen3-14B|
> |-|-|-|-|-|
> |S-GRPO|37.6/9,813|44.1/9,413|63.3/13079|63.8/12,419|
> |THoughtFold|39.1/9,102|46.9/8,713|65.5/11,670|69.7/11,217|
> ## Response to Weakness3:
> > All training uses math data, yet the paper claims generalization to GPQA without discussing whether the efficiency gains come from learned reasoning patterns or domain-specific shortcuts that happen to transfer.
>
> Thank you for this insightful comment. Our analysis (Sec. 4.3) shows that ThoughtFold learns generalized efficient reasoning patterns: concept graphs (Fig. 4) reveal reduced concept looping rather than mere domain-specific shortcuts, which aligns with our GPQA generalization. Admittedly, Fig. 5 confirms that domain-specific gains, such as sharper mathematical intuition, also contribute to overall improvement.
>
> ## Response to Question2 about computation overhead:
> > How many additional forward passes does this add per training step in practice, and how does wall-clock training time compare to standard GRPO?
>
> We use Qwen3-14B, the largest model in our experiments, for computational cost analysis.  Under identical settings and steps, ThoughtFold takes 23.53h vs. 21.44h for GRPO. While ThoughtFold introduces early rollout costs—requiring 6 search-and-verifications per correct response, with 4 parallel forward passes for answer tokens—its finer-grained supervision rapidly improves reasoning efficiency, which in turn speeds up rollout during training, making the overall wall-clock overhead acceptable.
>
> Moreover, LLM reasoning is an inference-intensive setting: under million-scale concurrent requests, a 40% efficiency gain over GRPO yields substantial savings, which can easily offset the extra ~2h of training cost. In addition, ThoughtFold also improves accuracy.
>
> ## Response to Question3:
> > Does removing non-contiguous steps during internal folding create artificial token sequences that allow the model to solve problems merely by pattern-matching the remaining fragments?
>
> We appreciate this insightful point. In practice, highly fragmented CoTs do not lead to simple pattern matching. Instead, the model typically reacts by initiating a second \<think\> phase or generating excessively long answer. To address this, our implementation filters out outputs with repeated think tags or abnormal lengths, marking them as incorrect even if the final answer is right. We will clarify this in the revision.
>
> ## Response to Question4:
> > Why use λ=0.1 when λ=0.01 yields better accuracy? Is this choice consistent across 4 models?
>
> In Table 3, λ=0.01 and λ=0.1 are two Pareto-optimal points. Compared with λ=0.01, λ=0.1 provides a substantial efficiency gain with only a small fluctuation in accuracy. λ=1.0 causes a clear accuracy drop. For efficient reasoning, λ=0.1 achieves the best trade-off between accuracy and efficiency, which is why we use it in paper. In practice, we believe both λ=0.01 and λ=0.1 can be useful under different application scenarios. We use the same setting across all four base models.
>
> **We sincerely thank the reviewer and hope these clarifications help address your concerns! If so, we would deeply appreciate it if you could consider raising your score.**
>
> [1] Does Representation Matter? Exploring Intermediate Layers in Large Language Models.NeurIPS2024
>
> [2] Layer by Layer: Uncovering Hidden Representations in Language Models.ICML2025

---

> > ### Author Rebuttal · Reviewer_zCHd · 2026-04-02
> >
> > Thanks a lot for your rebuttal, I would like to keep my score.

---

> > > ### Author Response · Authors · 2026-04-02
> > >
> > > Dear reviewer zCHd:
> > >
> > > Thank you for your kind support! We sincerely appreciate your valuable suggestions.
> > >
> > > With gratitude,
> > >
> > > Authors

---

### Official Review · Reviewer_G7QY · 2026-03-18

**Soundness:** 4
**Presentation:** 4
**Significance:** 3
**Originality:** 3
**Overall Recommendation:** 5
**Confidence:** 4

**Summary:**

This paper aims to improve the efficiency of reasoning paths in reasoning-oriented LLMs. Essentially after a correct trajectory is identified they go back and try to identify redundant or useless explorations and aim to remove them in the process of learning. This leads to significant reduction in token usage without concomitant decrease in quality.

**Compliance With Llm Reviewing Policy:**

Affirmed.

**Key Questions For Authors:**

- Please see weaknesses above.

**Limitations:**

yes

**Strengths And Weaknesses:**

Strengths
+ clear contribution reg long paths and how to truncate them in the process of learning
+ the empirical results are very strong and highlight that this approach works across model families and across different benchmarks.
+ the ablation results are very compelling. Although I didn't fully follow Table 2 it looks like masking is very important in the overall pipeline.

Weaknesses
- While masking appears quite important the actual masking and identifying which parts of the reasoning chain can be curtailed appears still a bit heuristic. For instance, do the steps outlined in Section 3 trap all the cases? For instance, what if you have a reasoning chain that looks almost redundant but it is because the chain is enumerating and trying out several possibilities one by one? (eg lets try x=2, x=3, etc. I imagine that will lead to a lot of redundancy in the chains but is not necessarily inefficient or wrong.)
- The actual way in which this approach updates the reasoning chains could be studied better in the experiments. In other words, what was the length of the original chain and what does the new one look like? How many loops got removed etc? Basically I am suggesting an analysis like Figure 4 but done over the full problem set to see in areas where the improvement happened what types of optimizations led to the improvement.

---

> ### Author Rebuttal · Authors · 2026-03-29
>
> Thank you for your valuable feedback and for recognizing our efforts! Following are our responses to each comment:
> ## Response to Weakness1:
> > While masking appears quite important the actual masking and identifying which parts of the reasoning chain can be curtailed appears still a bit heuristic. For instance, do the steps outlined in Section 3 trap all the cases? For instance, what if you have a reasoning chain that looks almost redundant but it is because the chain is enumerating and trying out several possibilities one by one? (eg lets try x=2, x=3, etc. I imagine that will lead to a lot of redundancy in the chains but is not necessarily inefficient or wrong.)
>
> Thank you for this insightful comment！
>
> ThoughtFold uses a simple principle to identify redundant steps: if removing certain steps still allows the model to follow a normal answer pattern and preserve accuracy, then those steps can be encouraged to be internalized and skipped. Dependency and redundancy are determined by the model's internal representations, rather than by manually specified rules. For example, an enumerative chain may appear redundant, but if removing it prevents correct reasoning, then it remains necessary; if the model can still reason correctly without it, then it is reasonable to encourage the model to skip it.
>
> Based on this principle, our design serves as a principled, practical strategy driven by the model's capabilities, rather than a rigid set of manual rules for every reasoning case, and the experiments and ablations provide empirical support for its effectiveness. As discussed in Sec. 3.2, masking is important because it avoids conflicting training signals on essential steps and enables precise supervision over shortcut transitions. We also acknowledge that the pruning and masking strategy still contains heuristic elements, and a more theoretical understanding of redundancy in reasoning chains is an important direction for future work.
> ## Response to Weakness2：
> > The actual way in which this approach updates the reasoning chains could be studied better in the experiments. In other words, what was the length of the original chain and what does the new one look like? How many loops got removed etc? Basically I am suggesting an analysis like Figure 4 but done over the full problem set to see in areas where the improvement happened what types of optimizations led to the improvement.
>
> Thank you for this valuable suggestion！
>
> Figure 4 provides a case study showing that ThoughtFold changes the model’s reasoning structure at a deeper level, rather than simply shortening the output. We agree that broader analyses of this kind would further strengthen the evidence. In fact, across the cases we have examined, we consistently observe the same trend, namely that ThoughtFold reduces redundant loops and restructures the reasoning process beyond surface-level shortening. Since this type of analysis is relatively expensive, we will add more examples covering problems of different difficulty levels in the revised version, and also report aggregate statistics such as the average reduction in loop-like patterns across datasets. We believe these additions will make the analysis more solid.
>
> **We sincerely thank the reviewer again and hope these clarifications help address your concerns!**

---

> > ### Author Rebuttal · Reviewer_G7QY · 2026-04-02
> >
> > Thanks for the response - I like the paper and recommend acceptance.

---

> > > ### Author Response · Authors · 2026-04-03
> > >
> > > Dear reviewer G7QY:
> > >
> > > Thank you for your kind support! We sincerely appreciate your valuable suggestions.
> > >
> > > With gratitude,
> > >
> > > Authors

---

### Official Review · Reviewer_Ttnk · 2026-03-19

**Soundness:** 2
**Presentation:** 3
**Significance:** 2
**Originality:** 2
**Overall Recommendation:** 2
**Confidence:** 4

**Summary:**

The paper tackles the "overthinking" problem in large reasoning models (LRMs) trained with RLVR . ThoughtFold proposes to identify redundant steps within correct chains using a two-phase introspective process (binary search on tail truncation + attention-based internal step pruning), then constructs step-level preference pairs from pruned vs. unpruned trajectories, and optimizes using a masked DPO loss alongside standard GRPO.

**Compliance With Llm Reviewing Policy:**

Affirmed.

**Ethical Review Concerns:**

The experiment results seem fabricated or mistakes. AIME has 30 problems, and the authors state in the paper that they do 16 random trials. Based on the total number of 480, no number of correct solutions will calculate accuracy that rounds to some of the reported results, e.g., no integer number of correct solutions can corresponds to the results of Qwen3-8B AIME2025 in table.1. e.g., 65.49*480/100 = 314.352, similarly, on AIME2024 the result of the proposed method was 57.2*480/100 = 274.56

**Ethical Review Flag:**

Flag this paper for an ethics review.

**Ethics Expertise Needed:**

["Research Integrity Issues (e.g., plagiarism)"]

**Final Justification:**

The most striking thing is the authors did not discuss nor cite OmegaPRM in a proper way. The authors admit in their rebuttal and the response seems merely specious arguing to me. OmegaPRM uses binary search to identify the first error step, and the authors uses binary search to identify the first redundancy step, clearly falling into the same family. The rebuttal argument that the approach is motivated to solve a different problem is valid, but this argument does not justify why the authors do not discuss OmegaPRM in the main paper -- and the authors clearly knew this work when writing this paper, yet they chose not to discuss in the paper.

Similarly, only after I asked about the detail of monte carlo rollout in binary search, the authors instantly claimed: "we use 4 parallel verifications and require an accuracy threshold of 0.75 for reliable preference pair construction." But this claim is conflicting with the algorithm C described in the appendix.

In particular, I was also suspicious about the results provided. The results do not seem realistic, but likely fabricated or mistakes. AIME has 30 problems, and the authors state in the paper that they do 16 random trials. Based on the total number of 480, no number of correct solutions will calculate accuracy that rounds to some of the reported results, e.g., no integer number of correct solutions can corresponds to the results of Qwen3-8B AIME2025 in table.1.

These specious arguing makes me concerned about the results, and more seriously, research integrity.

To be responsible, I maintain reject.

**Key Questions For Authors:**

1. How many verification samples do you use per pruning candidate in the binary search? If it's a single sample, have you measured how often the pruning decision is "wrong" (i.e., a pruned prefix succeeds by chance or a valid pruned prefix fails by chance)? How sensitive are your final results to using, say, 4 or 8 verification samples instead? This directly affects whether I trust the quality of your preference pairs.


2. Why not compare against SFT on shorter demonstrations (e.g., distilling from R1 into shorter CoTs via rejection sampling of short-but-correct trajectories)? This is arguably the simplest baseline for "make reasoning shorter" and its absence feels like an oversight. If ThoughtFold doesn't clearly beat this simple approach, the contribution is significantly diminished.

**Limitations:**

The authors is recommended to  discuss: (1) the fragility of single-sample verification in the pruning procedure, (2) the risk of breaking logical dependencies when pruning steps independently

**Strengths And Weaknesses:**

**Strengths:**

- The paper is generally well-written and the framework is presented in a logical, easy-to-follow manner.

- The two-phase pruning design — first chopping the tail, then folding internally — is intuitive and makes practical sense. It's a reasonable engineering solution.


**Weaknesses:**

*Soundness:*

- I have real concerns about the single-sample verification in the binary search procedure. The underlying assumption is that a prefix fails to get the correct answer because the prefix is over-pruned. This is not true, reasoning models are verbose and stochastic generators, the prefix likely fails to get the correct answer not because of over-pruned, but due to many other reasons, for example, the problem itself is difficult so the model rarely gets the right answer. Preference pairs constructed as such will be very noisy, many "chosen" responses are indeed suboptimal pruned reasoning traces.

- attention as importance: there seems insufficient justification for *why this works* or *which layer* or *how sensitive this is* feels underbaked.

- Steps are pruned independently based on importance scores, but reasoning steps have dependencies. If step 5 depends on step 3, and step 3 has low attention from the answer tokens, you'll remove step 3 and keep step 5 — which now makes no sense. The paper doesn't address this.

- No standard deviations or confidence intervals anywhere in Table 1. For AIME with 30 problems and 16 trials, the variance in accuracy could easily be several percentage points. The differences being claimed (e.g., 39.1% vs 35.1%) might be noise. I can't evaluate the empirical claims without this information.

*Presentation:*

- The "folding" metaphor is a bit strained. What's actually happening is pruning + preference learning. "Folding" sounds more sophisticated than what the method does. Not a dealbreaker, but the framing slightly oversells the conceptual contribution.

*Significance:*

- The results are incremental, not strong enough to convince readers why this particular approach matters more than alternatives. On MATH-500 (the easiest benchmark), accuracy actually drops slightly in some configurations. The gains on AIME are within what I'd expect from noise without variance bars. The real win is token reduction — but then, is this better than just training on shorter demonstrations via SFT? That comparison is conspicuously absent.

*Originality:*

- The individual components are all well-known: binary search idea is also seen in OmegaPRM,  masked DPO (from Gu et al. 2025). The combination is reasonable but not particularly surprising. I've seen similar prune-and-verify pipelines in other contexts. The paper reads more like solid engineering than a conceptual advance.

- Missing comparison with process reward models is a notable gap. PRMs do exactly what this paper is trying to do — assign step-level credit — and they're not even discussed as baselines, just briefly mentioned in limitations as "computationally expensive."

* Missing Discussion of OmegaPRM *
After a second review, I realize that the paper seem to miss the discussion of OmegaPRM, which I find it hard to overlook and have flag it. OmegaPRM has been around since June 2024 — it's a well-known, widely-cited approach that uses binary search to identify erroneous steps in reasoning traces. ThoughtFold's core procedure is clearly in the same family of ideas -- using binary search to identify a location where redundancy starts. However, the paper neither discusses OmegaPRM nor cites it or any of its follow-up works in any meaningful way. For anyone working in this space, this is going to stand out immediately. At minimum, the paper needs to situate itself properly relative to this line of work.

---

> ### Author Rebuttal · Authors · 2026-03-29
>
> Responses to each comment (W for Weakness, Q for Question):
> # W1&Q1 about Sounds:
> > Concerns about single-sample verification. Preference is noisy: a prefix can fail not because it is over-pruned.  Results under various verification samples?
>
> Verification is not single-sample: we use 4 parallel verifications and require an accuracy threshold of 0.75 for reliable preference pair construction. We will add this in the revision.
>
> We agree that a prefix is not necessarily incorrect even if all verifications fail. Our dynamic masking strategy addresses this challenge (see Sec.3.2) by avoiding penalties on correct prefix content and instead penalizing only shortcut transitions (e.g., directly answering). This is a key advantage of Fold concept: supervision is applied to transition probabilities between steps.
>
> Overall performance on Qwen3-8B with K verification samples, showing that our method is robust:
> |K|1|4|8|
> |-|-|-|-|
> |ACC|78.83|79.01|78.91|
> |Tokens|5901|5874 |5883|
> # W2 about Soundness:
> > attention as importance: why this works?which layer?how sensitive?
> - Why this works? The intuition is that the model implicitly distinguishes useful reasoning steps from redundancy, since final answer is concise, while reasoning chain is much longer. Thus, it is natural to measure step importance by its contribution to answer generation. Prior work (e.g.,[1]) shows that attention is effective proxy for token importance in LLM inference. Recent analysis [2] finds that, during answer generation in LRMs, attention over reasoning steps is sparse: a small subset of steps receive most attention. These results support attention as a reasonable proxy.
> - Which layer? We use middle-layer attention (see Line308). This choice is motivated by [3,4], which suggest shallow-layer representations are relatively noisy, while intermediate layers are more informative than deeper layers and provide better embeddings.
> - How sensitive? A layer ablation on Qwen3-8B shows that middle layer performs best and other choices degrade mildly:
> |Layer|First|Middle|Last|Avg|
> |-|-|-|-|-
> |ACC|78.72|79.01|78.85|78.91
> |Tokens|5899|5874|5911|5893
> # W3 about Soundness:
> > If step 5 depends on step 3, and 3 has low attention from answer, removing 3 and keeping 5 makes no sense.
>
> ThoughtFold identifies redundancy via a simple principle: if removing a step still preserves correctness, that step can be internalized. Dependency is determined by the model's internal representations, not by a manual requirement that Step 5 must explicitly rely on 3. Encouraging to remove such redundant local dependencies is exactly a feature of ThoughtFold.
> # W4 about Soundness:
> > No standard deviations.
>
> ACC with standard deviations for R1-distill-Qwen7B:
> ||Vanilla|Short-RL|Ours|
> |-|-|-|-|
> |AIME24|55.4±4.2|53.8±4.5|57.2±3.9
> |AIME25|39.1±2.6|35.1±2.9|39.1±2.4
>
> We will include all results in the revision.
> # W about Presentation&Originality:
> >  “Folding” is a bit strained. The individual components are well-known. No comparison with PRMs.
>
> Our motivation is to encourage the model to bypass redundant steps and directly connect essential reasoning segments. This is the intuition of FOLD.
>
> Built on this idea, we introduce an introspective strategy to identify redundancy in valid responses. Binary search is only used as a practical implementation. Another key component is the fold-anchor-based dynamic masking strategy, which enables precise preference supervision over shortcut probabilities between steps.
>
> ThoughtFold's core innovation is not step-level credit assignment, but learning shortcut preferences between reasoning steps, which has not been explored in prior work, including PRMs. We do not use PRMs as baseline because they have not been used for efficient reasoning (to our knowledge), and recent work [5] suggests that PRMs are unstable and limited in accuracy, making them unsuitable for this task.
> # W&Q2 about Significance:
> > The results are incremental. Why not compare against SFT?
>
> Our gains are not incremental: across 5 benchmarks and 4 models, we consistently improve accuracy and efficiency. Variance results above show these gains are robust. Fig.3–5 show that ThoughtFold changes reasoning structure rather than merely shortening output (noted by R-zCHd).
>
> S-GRPO is prior SOTA for efficient reasoning and already shows that SFT is clearly weaker than RL. Thus, we do not include a direct SFT baseline. Results for reference:
> |R1-distill-Qwen7B|AIME2024|MATH-500
> |-|-|-
> |ConCISE-SFT|52.1/9,751 |92.0/2,244
> |Ours|57.2/7,013|94.4/2,089
>
> [1] H2O: Heavy-Hitter Oracle for Efficient Generative Inference of Large Language Models. NeurIPS2023
>
> [2] FROST: Filtering reasoning outliers with attention for efficient reasoning. ICLR2026.
>
> [3] Does Representation Matter? Exploring Intermediate Layers in Large Language Models. NeurIPS2024
>
> [4] Layer by Layer: Uncovering Hidden Representations in Language Models. ICML2025
>
> [5] Promoting Efficient Reasoning with Verifiable Stepwise Reward. AAAI2026.

---

> > ### Author Rebuttal · Reviewer_Ttnk · 2026-04-04
> >
> > After a re-evaluation, I have to restate my concerns.
> >
> > **1. Missing Discussion of OmegaPRM**
> >
> > After a second review, I realize that the paper seem to miss the discussion of OmegaPRM, which I find it hard to overlook and have flag it. OmegaPRM has been around since June 2024 — it's a well-known, widely-cited approach that uses binary search to identify erroneous steps in reasoning traces. ThoughtFold's core procedure is clearly in the same family of ideas -- using binary search to identify a location where redundancy starts. However, the paper neither discusses OmegaPRM nor cites it or any of its follow-up works in any meaningful way. For anyone working in this space, this is going to stand out immediately. At minimum, the paper needs to situate itself properly relative to this line of work.
> >
> > And this might explain why other reviewers praise the originality of this work while I find it quite incremental
> >
> > ---
> >
> > **2. The Verification Rollout Issue — My Biggest Concern**
> >
> > This is where I have to be direct. Anyone who has worked with OmegaPRM, or really any binary-search-based step identification method, knows that the number of Monte Carlo rollouts used for verification is not a minor implementation detail — it's a core design choice that fundamentally affects the quality of everything downstream. So I was genuinely surprised to find no discussion of this anywhere in the paper.
> >
> > What makes this worse is that Algorithm 1 in Appendix C is written in a way that strongly implies a single rollout per verification step. If the authors had actually been using multiple rollouts, I would expect the algorithm to reflect that — it's the kind of thing you don't accidentally omit if it's part of your actual procedure.
> >
> > It was only after I asked directly that the authors mentioned: *"we use 4 parallel verifications and require an accuracy threshold of 0.75 for reliable preference pair construction."* That's a meaningful detail. But it never appeared in the paper. For a hyperparameter this consequential to reproducibility, that's not acceptable. I want to be fair to the authors here — maybe this really was an oversight rather than more serious thing like research integrity— but I can't just take that on faith when the algorithm as written tells a different story. This is the kind of thing that makes it hard to fully trust the reported results.
> >
> > ---
> >
> > **3. Incremental Gains, No Statistical properties reported**
> >
> > Stepping back to look at the results more broadly: compared to prior state-of-the-art, ThoughtFold achieves roughly another 5% reduction in token count. That's fine, but it's a modest gain, and I'd want to see it backed up with proper statistical reporting, such as standard deviations. Are the authors running multiple samplings for each question? Are the results statistically reliable?
> >
> > This matters a lot here. Long chain-of-thought reasoning models are notoriously high-variance, a few percentage points of accuracy difference can easily be noise. And this concern feeds directly back into the rollout issue: if the preference pairs are being constructed with noisy single-rollout verifications (as the algorithm implies), then the pruned traces themselves will be noisy, and the results built on top of them become harder to interpret.
> >
> > The authors say they use four parallel verifications — and maybe they do — but given that this detail was absent from the paper entirely until I asked, I'm not in a position to simply accept that and move on.
> >
> > ---
> >
> > I want to be clear: I'm raising these concerns because I'm familiar with this line of work — particularly OmegaPRM and the broader binary-search-based verification literature — and that familiarity is precisely why some of these omissions stand out to me as unusual. The missing rollout detail, the mismatch between the algorithm as written and the authors' post-hoc clarification, and the lack of any statistical uncertainty reporting are, taken together, things I think the AC would want to know when making their decision. I genuinely hope I'm wrong, but I don't think it would be responsible for me to stay quiet about it.

---

> > > ### Author Response · Authors · 2026-04-05
> > >
> > > Thank you for your valuable comments.
> > > ## 1. Discussion of OmegaPRM
> > > We already clarified in our first-round rebuttal that PRM methods, including OmegaPRM [1], are not the right fit for our setting. Here we further clarify that **OmegaPRM and ThoughtFold differ substantially and fundamentally in both application scenario and core technique:**
> > > - OmegaPRM
> > >   - Setting: PRM data construction.
> > >   - Technique: a PRM data construction framework built on MCTS, where binary search is used to efficiently locate the first error.
> > > - ThoughtFold
> > >   - Setting: reducing redundancy in LRM efficient reasoning.
> > >   - Technique: an Introspective Preference Learning framework, where prune-and-verify is used to identify redundancy (binary search serves only as an efficient search strategy) and dynamic masking enables precise preference construction to encourage the model to bypass redundancy and bridge essential reasoning parts.
> > >
> > > **In summary**, ThoughtFold is an RL framework that encourages the model to bypass redundancy and directly bridge essential reasoning parts for efficient reasoning, which is the core innovation of our work. OmegaPRM is an automated process-supervision data construction framework for training PRMs. **As we already clarified in our first rebuttal, our key novelty relative to prior work is not step-level credit assignment, but preference construction over transitions between reasoning steps.**
> > >
> > > **Regarding binary search, it is a simple and efficient classical search algorithm.** ThoughtFold uses binary search in prune-and-verify process as an efficient search strategy for identifying redundancy, whereas OmegaPRM uses it to locate first-error. Binary search is neither our innovation nor OmegaPRM’s innovation. It is simply an efficient existing module within the respective framework. Moreover, our internal folding is fundamentally different in form and purpose from OmegaPRM’s first-error localization.
> > >
> > > Although OmegaPRM is proposed in a rather different setting, we agree that there is some similarity in search strategy. We will **add a discussion in the Appendix** of recent search-based process-supervision works related to OmegaPRM[1], such as EpicPRM[2] and STEPCO[3]:
> > > - Their goal is to generate step-level supervision for PRM training, while ours is not a supervision or credit assignment method. ThoughtFold differs in both objective and mechanism: it uses model introspection to identify redundancy, and mask-based preference learning to encourage the model to fold them by bypassing redundant steps and directly connecting the essential reasoning parts.
> > >
> > > ## 2. Verification Rollout
> > > - This is a **minor implementation detail** regarding how we verify whether the model can derive the correct answer from the pruned CoT. We therefore omitted it in the submitted version. As we recognized during the first-round discussion, this is a detail that readers may reasonably care about, and we will add it to the implementation details in the revision. Concretely, we use 4 parallel verification rollouts.
> > > - We **have included an ablation study** on this important parameter in the first-round rebuttal. The results show that reducing the number of verification samples to 1 causes only slight performance fluctuations, without fundamentally affecting training stability or the final outcome.
> > > - We will **release the code upon acceptance** to support reproducibility.
> > >
> > > **Overall, we sincerely thank you for raising this point, which helps us further improve the technical clarity and implementation details of the paper.**
> > >
> > > ## 3. No Statistical properties
> > > In fact, **all of these concerns were already clarified in either the paper or our first-round rebuttal:**
> > > - In Sec. 4.1 (Metrics) of the paper, we explicitly state that, at test time, we follow the sampling setup of S-GRPO [4] (e.g., 16 samples per question on AIME).
> > > - We have reported the standard deviations in the first-round rebuttal and will include it in the revision.
> > > - Our method improves not only token efficiency, but also accuracy by a substantial margin. Regarding whether the gains are merely incremental, we already provided a detailed analysis in the first-round rebuttal, showing that our method can fundamentally reshape the model’s reasoning structure (see also Sec. 4.3 of the paper and the comments from R-zCHd).
> > >
> > > **We sincerely appreciate the reviewer’s time and many thoughtful comments, which have helped us better refine and strengthen our work! We hope that our responses have adequately addressed your concerns. If so, we would deeply appreciate it if you could consider raising your score.**
> > >
> > > [1] Improve Mathematical Reasoning in LanguageModels by Automated Process Supervision.
> > >
> > > [2] An Efficient and Precise Training Data Construction Framework for Process-supervised Reward Model in Mathematical Reasoning. ACL2025
> > >
> > > [3] Enhancing Mathematical Reasoning in LLMs by Stepwise Correction. ACL2025
> > >
> > > [4] S-GRPO: Early Exit via Reinforcement Learning in Reasoning Models. NeurIPS2025

---

### Official Review · Reviewer_Xndy · 2026-03-22

**Soundness:** 3
**Presentation:** 3
**Significance:** 3
**Originality:** 3
**Overall Recommendation:** 5
**Confidence:** 4

**Summary:**

This paper proposes a novel framework (ThoughtFold) that integrates outcome-based RLVR with fine-grained preference learning​ to explicitly penalize redundant explorations and enhance efficient reasoning. The core designs of introspective preference learning include two parts:(i) Preference Data Construction between Concise Success and Over-Simplified Failure. Specifically, the Introspective strategy is used to distinguish the above two types of samples, which is based on Tail Truncation(self-repetition) and Internal Folding (off-target attempts) through step-level importance scores (model's attention weights); (ii)  Dynamic Mask Strategy integrated with the original DPO:  For each preference pair, it constructs step-level binary masks to apply precise supervision for different preferrences. Through combining the standard GRPO optimization with the fine-grained Mask DPO strategy, ThoughtFold can further enhance efficient and effective reasoning than previous methods across various reasoning benchmarks.

**Compliance With Llm Reviewing Policy:**

Affirmed.

**Final Justification:**

The paper presents a novel, well-motivated approach to mitigate overthinking in reasoning models via introspective preference learning and attention-guided step pruning. It demonstrates strong originality, clear writing, and convincing experimental validation.

**The authors’ rebuttal comprehensively addresses both major concerns**:

1. They provide concrete **training time overhead** comparisons and clarify that hyperparameter tuning is minimal and stable.
2. They supply in-depth justification for **attention as importance proxy**, grounded in prior theory and empirical sparsity, with layer ablation proving robustness.

Based on the above evaluation, I am glad to recommend acceptance of this paper.

**Key Questions For Authors:**

Please see the above Weaknesses.

**Limitations:**

yes

**Strengths And Weaknesses:**

**Strengths**

**(1) Clear Motivation and Problem Formulation:** The paper clearly identifies the core issue of "overthinking" in current Large Reasoning Models trained with Reinforcement Learning with Verifiable Rewards. It correctly attributes this to the coarse-grained, outcome-only reward signal in RLVR, which causes models to memorize and reinforce redundant explorations within correct reasoning trajectories.

**(2) Well-Organized Method:** The clarification of introspective preference learning is clear and reasonable in some way. Moreover, the design of the step-level importance scores (from the models' attention weights) is interesting.

**(3) Conving Experimental Results:** Extensive experiments are conducted across different benchmarks(GSM8K, MATH-500, AIME, GPQA) on different backbones. The performance and efficiency gain can verify the effectivenss the proposed method.

**Weaknesses**

**(1) Lack of Discussions about Computational Overhead:** ThoughtFold introduces multiple additional components, including introspective search, attention computation, and dynamic mask construction, making the overall training pipeline more complex compared to standard GRPO or simple length-penalty RL. Although the paper notes that the introspective process incurs negligible overhead as attention is computed only once per correct sample, the overall training implementation and hyperparameter tuning (e.g., for the balancing coefficient λ) complexity is increased.

**(2) Lack of Theoretical Analysis for the effectiveness of Attention Weight:** The Internal Folding phase relies on attention weights to achieve "importance." The ablation study can verify that attention guidance is superior to a random policy. However, the paper does not discuss whether attention is always the reliable proxy for redundancy. More detailed Theoretical Analysis are preferred.

---

> ### Author Rebuttal · Authors · 2026-03-29
>
> Thank you for your valuable feedback and for recognizing our efforts! Following are our responses to each comment:
> ## Response to Weakness1:
> > ThoughtFold introduces multiple additional components, including introspective search, attention computation, and dynamic mask construction, making the overall training pipeline more complex compared to standard GRPO or simple length-penalty RL. Although the paper notes that the introspective process incurs negligible overhead as attention is computed only once per correct sample, the overall training implementation and hyperparameter tuning (e.g., for the balancing coefficient λ) complexity is increased.
>
> To assess computational cost, we report results on Qwen3-14B, the largest model used in our experiments. Under the same training setting and number of steps, Short-RL takes 17.49h, ThoughtFold 23.53h, and GRPO 21.44h. ThoughtFold introduces some rollout costs early in training, but its finer-grained supervision improves efficiency faster than Length Penalty, making the overall overhead acceptable.
> Importantly, ThoughtFold does not introduce prohibitive tuning complexity. The core mechanism—encouraging the model to bypass redundancy—is conceptually straightforward. As a result, the primary tuning factor is the coefficient $\lambda$, which offers a predictable and controllable trade-off between accuracy and efficiency (as shown in Table 3). Beyond this, ThoughtFold requires minimal additional hyperparameter engineering compared to standard GRPO.
> ## Response to Weakness2:
> > The Internal Folding phase relies on attention weights to achieve "importance." The ablation study can verify that attention guidance is superior to a random policy. However, the paper does not discuss whether attention is always the reliable proxy for redundancy. More detailed Theoretical Analysis are preferred.
>
> Thank you for this valuable comment! We provide a deeper justification of our attention-based proxy here (you can refer to my responses to Reviewer-NRrv and Reviewer-Ttnk for more details):
> - **Why attention?** The intuition is that the model implicitly distinguishes useful reasoning steps from redundancy, since the final answer is concise, while the reasoning chain is longer and redundant. Thus, it is natural to measure the step importance by its contribution to answer generation. The choice of attention is supported by prior work (e.g., [1]), which shows that attention scores are effective proxies for token importance in LLM inference and provide **theoretical justification**. Moreover, recent analysis [2] finds that, during answer generation in LRMs, attention over reasoning steps is highly sparse: a small subset of steps receives most of the attention, while many others receive almost none. This suggests that using attention scores to identify redundant steps is reasonable and practical.
> - **Why middle-layer attention?** We use middle-layer attention. This choice is motivated by prior studies [3,4], which suggest shallow-layer representations are relatively noisy, while intermediate layers are more informative than deeper layers and provide better embeddings.
> - **How sensitive?**  Our layer ablation on Qwen3-8B shows the middle layer performs best, with other choices degrading only mildly. This confirms ThoughtFold's robustness to layer selection and validates attention as a stable importance proxy, as detailed below:
> |Layer|First|Middle|Last|All-Layer-Avg|
> |-|-|-|-|:-:|
> |ACC|78.72|79.01|78.85|78.91|
> |Tokens|5899|5874|5911|5893|
>
> We will include the above analysis in the next version.
>
> We also agree that a more rigorous theoretical analysis of attention-based selection in reasoning remains a valuable open question. Your perspective is truly forward-looking, and we view this as a significant direction for future research.
>
> **We sincerely thank the reviewer again and hope these clarifications help address your concerns!**
>
> [1] H2O: Heavy-Hitter Oracle for Efficient Generative Inference of Large Language Models.NeurIPS2023
>
> [2] FROST: Filtering reasoning outliers with attention for efficient reasoning.ICLR2026.
>
> [3] Does Representation Matter? Exploring Intermediate Layers in Large Language Models.NeurIPS2024
>
> [4] Layer by Layer: Uncovering Hidden Representations in Language Models.ICML2025

---

> > ### Author Rebuttal · Reviewer_Xndy · 2026-04-01
> >
> > Thanks for your detailed replies that address most of my concerns. I have increased the score and hope you can integrate the explanations in your revised version!

---

> > > ### Author Response · Authors · 2026-04-01
> > >
> > > Dear reviewer Xndy:
> > >
> > > Thank you for your positive assessment of our work and your decision to increase the score from 4 to 5. We are truly grateful for your insightful suggestions, and will integrate the relevant points from our discussion into the revised version of the manuscript.
> > >
> > > With gratitude,
> > >
> > > Authors

---

### Official Review · Reviewer_NRrv · 2026-03-23

**Soundness:** 3
**Presentation:** 3
**Significance:** 2
**Originality:** 3
**Overall Recommendation:** 4
**Confidence:** 4

**Summary:**

This paper introduces ThoughtFold, a post-training method that encourages models to produce shorter chains of thought for reasoning tasks while preserving performance. The approach builds on GRPO and adds a step-level regularization signal that penalizes redundant exploration within correct reasoning trajectories. Experiments show an improved efficiency-accuracy trade-off on mid sized LLMs.

**Compliance With Llm Reviewing Policy:**

Affirmed.

**Key Questions For Authors:**

1. You show some average token length for pass@k, but what about accuracy?
2. See Weaknesses: what are the ACTUAL computational costs?

**Limitations:**

Again, the computational cost needs to be spelled out, compared to simple methods like length regularization.

**Strengths And Weaknesses:**

Strengths:
1. The method proceeds with what one probably would do first: Truncate, then check whether the model can still output the right results and if so truncate more. Then remove chunks that don't seem to matter. But using contrastive pairs (DPO-style) is a cute idea.
2. Results show that it work, though one could argue that it's close to other length aware training methods.

Weaknesses:
1. The chunking into steps seems rather arbitrary and hardcoded (like "\n") and not really justified
2. More seriously, a real discussion of the computational overhead is lacking beyond "minimal computational overhead"). This would be important because the performance (i.e. token-length) difference to the other length-minimizing SOTA methids is not huge, so we need to see real comparisons along the compute axis (length penalty strikes me as much cheaper).
3. The method is a brew of several ideas thrown together which makes it less appealing and also less convincing. Ablations seem to show you need every little piece of, which looks somewhat like hillclimbing.
4. The ranking with average attention to the target seems complicated and moreover it is never discussed what layer is used (you say middle layer, but which and why not any other? Seems like another knob to hillclimb).

---

> ### Author Rebuttal · Authors · 2026-03-29
>
> Thank you for your thoughtful comment! Following are our responses to each comment.
> ## W1:
> > The chunking into steps seems rather arbitrary and hardcoded (like "\n") and not justified
>
> We follow the standard criterion ("\n\n") for reasoning-step segmentation and most prior work adopts the same convention (e.g., [1-4]).
> ## W2&Q2:
> > What are the ACTUAL computational costs?
>
> We use Qwen3-14B, the largest model in our experiments, for computational cost analysis. Under the same training setting and number of training steps, Short-RL takes 17.49h, GRPO takes 21.44h, and ThoughtFold takes 23.53h. ThoughtFold does incur some additional rollout cost early in training, but its finer-grained supervision improves reasoning efficiency faster than Length Penalty, making overall training overhead acceptable.
>
> More importantly, LLM reasoning is an **inference-intensive** setting: under million-scale concurrent requests, a 4% efficiency gain over Length Penalty can translate into substantial savings, which can easily offset the extra ~6h of training cost. Moreover, Length Penalty causes a substantial accuracy drop, especially on hard benchmarks AIME24&25, where it is 4.0% and 6.6% below ThoughtFold, making it less reliable.
> ## W3 about novelty:
> > The method is a brew of several ideas thrown together which makes it less appealing and also less convincing. Ablations seem to show you need every little piece of, which looks somewhat like hillclimbing.
>
> Our core innovation is to encourage the model to bypass redundant steps and directly connect essential reasoning segments by constructing preference signals over step transitions.
>
> Each component of ThoughtFold is essential to this objective: the introspective strategy identifies redundancy, and the dynamic masking strategy avoids conflicting credit assignment and enables precise supervision over shortcut transitions. Together, they precisely penalize redundant parts and encourage the model to directly bridge essential reasoning steps.
>
>  The ablation further supports our design. For example, removing dynamic masking causes a performance drop.
> ## W4:
> > The ranking with average attention to the target seems complicated and moreover it is never discussed what layer is used (you say middle layer, but which and why not any other? Seems like another knob to hillclimb).
>
> - **Why attention?** The intuition is that the model implicitly distinguishes useful steps from redundancy, since final answer is concise, while reasoning chain is much longer. It is therefore natural to measure step importance by its contribution to answer generation. Prior work (e.g., [5]) demonstrates that attention scores are effective proxies for token importance in LLM inference. Recent analysis [6] further finds that, during answer generation in LRMs, attention over reasoning steps is highly sparse, with only a small subset of steps receiving most of the attention. These results support attention as a principled and practical importance proxy.
>
> - **Which layer?** Prior studies [7,8] suggest that shallow-layer representations are relatively noisy, while intermediate layers are more informative than deeper layers and provide better embeddings. We therefore use middle-layer attention.
>
> - **Sensitivity analysis.** We conduct a layer ablation on Qwen3-8b. The middle layer performs best, while the other choices degrade only mildly. This shows ThoughtFold is robust to layer choice and attention is a stable importance proxy. We report the results below.
> |Layer|First|Middle|Last|All-Layer-Avg
> |-|-|-|-|:-:
> |ACC|78.72|79.01|78.85|78.91
> |Tokens|5899|5874|5911|5893
> ## Q1:
> > You show some average token length for pass@k, but what about accuracy?
>
> We introduce min length@k to analyze whether ThoughtFold changes the model’s reasoning structure rather than merely shortening outputs (see Section 4.3).
> We report the pass@k results of AIME24 on Qwen3-14B below.
> |k|1|4|8|16
> |-|-|-|-|-
> |Vanilla |75.4| 84.6|88.2|90.0
> |Short-RL|75.2|80.8|83.3|86.7
> |Ours|79.2|88.1|91.6|93.3
>
> **We will include the above analyses in the revision. We sincerely thank the reviewer and hope our clarifications help address your concerns! If so, we would deeply appreciate it if you could consider raising your score.**
>
> [1]Step-by-Step Reasoning for Math Problems via Twisted Sequential Monte Carlo.ICLR2025.
>
> [2]Can Large Language Models Detect Errors in Long Chain-of-Thought Reasoning?ACL2025
>
> [3]Thinking-Free Policy Initialization Makes Distilled Reasoning Models More Effective and Efficient Reasoners.ICLR2026
>
> [4]Causal Sufficiency and Necessity Improves Chain-of-Thought Reasoning.NeurIPS2025
>
> [5]H2O: Heavy-Hitter Oracle for Efficient Generative Inference of Large Language Models.NeurIPS2023
>
> [6]FROST: Filtering reasoning outliers with attention for efficient reasoning.ICLR2026
>
> [7]Layer by Layer: Uncovering Hidden Representations in Language Models.ICML2025
>
> [8]Does Representation Matter? Exploring Intermediate Layers in Large Language Models.NeurIPS2024

---

> > ### Author Rebuttal · Reviewer_NRrv · 2026-03-31
> >
> > You answered my questions and I hope you will include some of this in the revision. I am raising my score by one notch.

---

> > > ### Author Response · Authors · 2026-04-01
> > >
> > > Dear reviewer NRrv:
> > >
> > > Thank you for your kind support and your decision to raise the score from 3 to 4. We sincerely appreciate your valuable suggestions. We will carefully incorporate the relevant points from our discussion into the revised manuscript.
> > >
> > > With gratitude,
> > >
> > > Authors

---

### Decision · Program_Chairs · 2026-04-30

**Decision:**

Accept (regular)

**Comment:**

This paper introduces a post-training method to solve "over-thinking" issues in existing large reasoning models. Reviewers admit that this paper is well-motivated and well-written, with solid experimental results. At the initial review stage, reviewers raise these concerns include:

- Reviewer NRrv: 1. computational overhead; 2. Combined works; 3. Discussion on Layers (Attention).
- Reviewer Xndy: 1. computational overhead; 2. Lack of Theoretical Analysis for Attention.
- Reviewer G7QY: 1. Details about identify reasoning part; 2. Study on how to update reasoning.
- Reviewer zCHd: 1. Discussion on Layers; 2. Missing results; 3. Only Math data.
- Reviewer Ttnk: 1. Verification in binary search; 2. Discussion on Layers (Attention); 3. Dependency on Reasoning steps; 4. No standard deviations; 5. incremental results; 6. Combined works; 7. Missing comparisons; 8. Miss discussion of OmegaPRM.

After the discussion stage, authors have provided details respone to address these concerns as:
1. computational overhead: Authors admit that ThoughtFold indeed incur some additional rollout at the training but can make LLM inference is efficiency.
2. combined works: Authors claim the importance of each module, that introspective strategy identifies redundancy, and the dynamic masking strategy avoids conflicting credit assignment and enables precise supervision over shortcut transitions.
3. Discusson on Attention Layers: Authors provide some related works to demonstrate the importance of these parts. (**I suggest authors should add these parts into the final version**)
4. Identify reasoning part and Study on updating reasoning: Authors have provided more clear details for these parts.
5. Missing results and Only Math data: Authors have explained why missing (lack code) and have compared them now.

So, for these responses, I think the minor issues is should add the discussion about attention layers into the final version as all reviewers raise such a concern. These discussion should not only at the rebuttal stage but also at the final stage. Besides, most of concerns from each reviewer except Ttnk have been addressed.

For reviewer Ttnk, he has pointed out this paper have missed citations of OmegaPRM and missing verifications steps, i.e., *we use 4 parallel verifications and require an accuracy threshold of 0.75 for reliable preference pair construction.* (authors agree this point and promise to add these details but also claim that only cause slight performance fluctuations).

From these discussion, I think from the proposed method and OmegaPRM have different motivation in solving tasks, but also should add some citations about it to explain the different to avoid misunderstanding. For other details like verifications is a missing details but author should add it though it does not matter the final method.